# Beyond the 80/20 Rule:
# High-Entropy Minority Tokens Drive Effective Reinforcement Learning for LLM Reasoning

**Shenzhi Wang♠,♣, Le Yu♣, Chang Gao♣, Chujie Zheng♣, Shixuan Liu♣, Rui Lu♠,**
**Kai Dang♣, Xiong-Hui Chen♣, Jianxin Yang♣, Zhenru Zhang♣, Yuqiong Liu♣, An Yang♣,**
**Andrew Zhao♠, Yang Yue♠, Shiji Song♠, Bowen Yu♣,✉,†, Gao Huang♠,✉, Junyang Lin♣**

♠ Department of Automation, BNRist, Tsinghua University    ♣ Qwen Team, Alibaba Inc.

## Abstract

Reinforcement Learning with Verifiable Rewards (RLVR) has emerged as a powerful approach to enhancing the reasoning capabilities of Large Language Models (LLMs), while its mechanisms are not yet well understood. In this work, we undertake a pioneering exploration of RLVR through the novel perspective of token entropy patterns, comprehensively analyzing how different tokens influence reasoning performance. By examining token entropy patterns in Chain-of-Thought (CoT) reasoning, we observe that only a small fraction of tokens exhibit high entropy, and these tokens act as critical *forks* that steer the model toward diverse reasoning pathways. Furthermore, studying how entropy patterns evolve during RLVR training reveals that RLVR largely adheres to the base models entropy patterns, primarily adjusting the entropy of high-entropy tokens. These findings highlight the significance of high-entropy tokens (i.e., forking tokens) to RLVR. We ultimately improve RLVR by restricting policy gradient updates to forking tokens and uncover a finding even beyond the 80/20 rule: utilizing only 20% of the tokens while maintaining performance comparable to full-gradient updates on the Qwen3-8B base model and significantly surpassing full-gradient updates on the Qwen3-32B (+11.04 on AIME'25 and +7.71 on AIME'24) and Qwen3-14B (+4.79 on AIME'25 and +5.21 on AIME'24) base models, highlighting a strong scaling trend. In contrast, training exclusively on the 80% lowest-entropy tokens leads to a marked decline in performance. These findings indicate that the efficacy of RLVR primarily arises from optimizing the high-entropy tokens that decide reasoning directions. Collectively, our results highlight the potential to understand RLVR through a token-entropy perspective and optimize RLVR by leveraging high-entropy minority tokens to further improve LLM reasoning.

## 1 Introduction

The reasoning capabilities of Large Language Models (LLMs) have advanced substantially in domains like mathematics and programming, propelled by test-time scaling methodologies employed in OpenAI o1 [24], Claude 3.7 [1], DeepSeek R1 [6], Kimi K1.5 [34], and Qwen3 [41]. A pivotal technique driving these improvements is Reinforcement Learning with Verifiable Rewards (RLVR) [17, 6, 41], where models optimize outputs through RL objectives tied to automated correctness verification. While recent RLVR advancements have stemmed from algorithmic innovations [42, 44, 10], cross-domain applications [40, 21, 26], and counterintuitive empirical in-

---

✉: Corresponding Authors    †: Project Lead

Emails:  `wangshenzhi99@gmail.com`  `yubowen.ph@gmail.com`  `gaohuang@tsinghua.edu.cn`

Project Page: `https://shenzhi-wang.github.io/high-entropy-minority-tokens-rlvr`

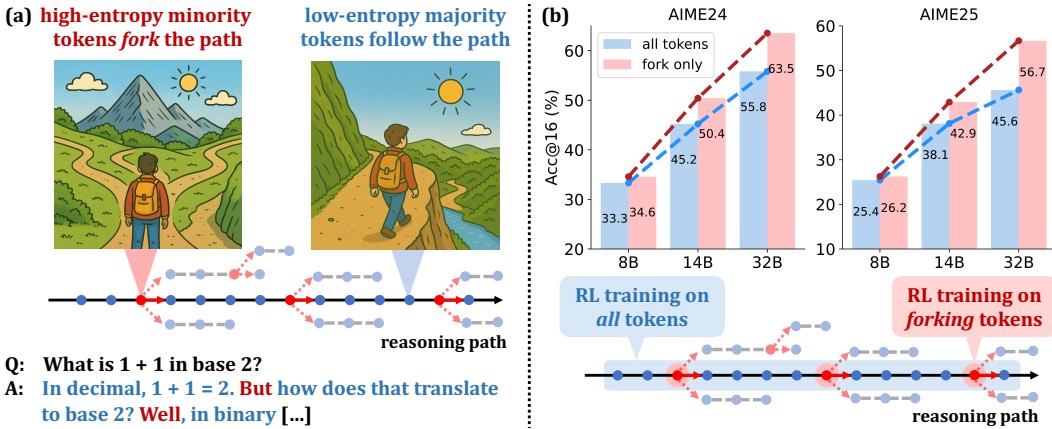

Figure 1: (a) In CoTs, only a minority of tokens exhibit high entropy and act as "forks" in reasoning paths, while majority tokens are low-entropy. (b) RLVR using policy gradients of forking tokens delivers significant performance gains that scale with model size. With a 20k maximum response length, our 32B model sets new SoTA scores (63.5 on AIME'24 and 56.7 on AIME'25) for RLVR on base models under 600B. Extending the maximum response length to 29k further boosts the AIME'24 score to 68.1.

sights [37, 43, 46], existing implementations directly train over all tokens with limited understanding of which tokens actually facilitate reasoning. These approaches neglect the heterogeneous functional roles tokens play in reasoning processes, potentially hindering further performance gains by failing to prioritize critical decision points in sequential reasoning trajectories.

In this paper, we analyze the underlying mechanisms of RLVR through an innovative lens of token entropy patterns, investigating how tokens with varying entropy impact reasoning performance. We first point out that in the Chain-of-Thought (CoT) processes of LLMs, the entropy distribution exhibits a distinct pattern where the majority of tokens are generated with low entropy, while a critical minority of tokens emerge with high entropy. Through comparing the textual meanings of these two parts of tokens, we observe that the tokens with lowest average entropy primarily complete the ongoing linguistic structures, while the tokens with highest average entropy function as pivotal decision points that determine the trajectory of reasoning among multiple potential pathways (referred to as *forks*), as depicted in Figure 1(a). In addition to qualitatively anslysis, we conduct controlled experiments by manually modulating the entropy of forking tokens during decoding. Quantitative results reveal that moderately increasing the entropy of these high-entropy forking tokens leads to measurable improvements in reasoning performance, while artificially reducing their entropy results in performance degradation, confirming the importance of maintaining high entropy and the role as "forks" for these high-entropy tokens. Furthermore, by analyzing the evolution of token entropy during RLVR training, we find that the reasoning model largely retains the entropy patterns of the base model, exhibiting only gradual and relatively minor changes as training progresses. Additionally, RLVR primarily changes the entropy of high-entropy tokens, while the entropy of low-entropy tokens varies only within a small range. The above observations highlight the critical role high-entropy minority tokens may play in CoTs and RLVR training.

Building upon the discovery of forking tokens, we further refine RLVR by retaining policy gradient updates for only 20% of tokens with the highest entropy and masking gradients of the remaining 80%. We observe that although solely utilizing 20% of tokens, our approach can still achieve competitive reasoning performance on Qwen3-8B base model compared to full-gradient updates. Moreover, its effectiveness increases with model size, yielding reasoning improvements of +11.04 on AIME'25 and +7.71 on AIME'24 for the Qwen3-32B base model, and +4.79 on AIME'25 and +5.21 on AIME'24 for the Qwen3-14B base model, as shown in Figure 1(b). Notably, the 32B model trained with only 20% high-entropy tokens attains scores of 63.5 on AIME'24 and 56.7 on AIME'25, setting a new state-of-the-art (SoTA) for reasoning models trained directly from base models with fewer than 600B parameters. Extending the maximum response length from 20k to 29k further elevates our 32B model's AIME'24 score from 63.5 to 68.1. Conversely, training exclusively on the 80% lowest-entropy tokens results in severe performance degradation. These observations show that 20% of tokens achieve performance comparable to or exceeding 100%, even surpassing

the 80/20 rule. The results demonstrate that the high-entropy minority tokens, functioning as critical decision points in reasoning trajectories, account for nearly all performance gains in RLVR.

Finally, we explore why retaining a small fraction of the highest-entropy tokens leads to strong performance in RLVR via a series of ablation studies. We adjust the chosen fraction of forking tokens, either by decreasing it from 20% to 10% or increasing it to 50% or 100%, and report the corresponding reasoning metrics and overall entropy during the training process. Experimental results demonstrate that retaining approximately 20% of the highest-entropy tokens optimally balances exploration and performance, while deviating from 20% reduces the overall entropy with diminished exploration and incurs worse performance. This suggests that only a critical subset of high-entropy tokens meaningfully contributes to the exploration during RL while others may be neutral or even detrimental. Reducing the proportion to 10% removes certain useful tokens, which weakens exploration. Increasing the proportion to 50% or 100% adds low-entropy tokens, which also reduces the effectiveness of exploration. Last but not least, retaining the top 20% of high-entropy tokens results in the largest performance gains for the 32B model, followed by the 14B model, and the smallest gains for the 8B model. This may be due to the insufficient capacity of the smaller model, which restricts its ability to benefit from increased exploration. These findings highlight the importance of preserving an appropriate proportion of high-entropy tokens in RLVR. As model size increases, the strategy of selecting high-entropy tokens appears to scale effectively.

In summary, our findings emphasize the pivotal role of high-entropy minority tokens in shaping the reasoning abilities of LLMs. We hope this inspires further analyses from the perspective of token entropy and informs more effective RLVR algorithms that strategically leverage these tokens to enhance reasoning performance. The key takeaways of our paper are as follows:

- In CoTs, the majority of tokens are generated with low entropy, while only a small subset exhibits high entropy. These high-entropy minority tokens often act as "forks" in the reasoning process, guiding the model toward diverse reasoning paths. Maintaining high entropy at these critical forking tokens is beneficial for reasoning performance. (§2)

- During RLVR training, the reasoning model largely preserves the base model's entropy patterns, showing only gradual and minor changes. RLVR primarily adjusts the entropy of high-entropy tokens, while the entropy of low-entropy tokens fluctuates only within a narrow range. (§3)

- High-entropy minority tokens drive nearly all reasoning performance gains during RLVR, whereas low-entropy majority tokens contribute little or may even hinder performance. One possible explanation is that, prior to performance convergence, a subset ($\sim 20\%$ in our experiments) of high-entropy tokens facilitates exploration, while low-entropy tokens offer minimal benefit or may even impede it. (§4)

- Based on the insights above, we further discuss (i) high-entropy minority tokens as a potential reason why supervised fine-tuning (SFT) memorizes but RL generalizes, (ii) how prior knowledge and readability requirements shape the different entropy patterns seen in LLM CoTs compared to traditional RL trajectories, and (iii) the advantage of clip-higher over entropy bonus for RLVR. (Deferred to Appendix D)

## 2    Analyzing Token Entropy in Chain-of-Thought Reasoning[1]

Although prior works [41, 42, 44] have highlighted the importance of generation entropy in chain-of-thought reasoning, they typically analyze the entropy of all tokens collectively. In this section, we take a closer look at generation entropy in chain-of-thought reasoning by examining it at the token level. To this end, we use Qwen3-8B [41], one of the most recent and capable reasoning models within a comparable parameter scale, to generate responses for queries from AIME'24 and AIME'25, using a decoding temperature of $T = 1.0$. We enforce the use of the thinking mode for every question and collect over $10^6$ response tokens. For each token, the entropy is computed according to the formulation in Equation (3). The statistical analysis of the entropy values of these $10^6$ tokens is presented in Figure 2. Furthermore, a visualization of token entropy for an entire long CoT response is provided in Figures 12 to 17 in the Appendix. From these analyses, we identify the following entropy patterns:

---

[1]Due to the page limit, the preliminaries and related work are deferred to Appendix A and B, respectively.

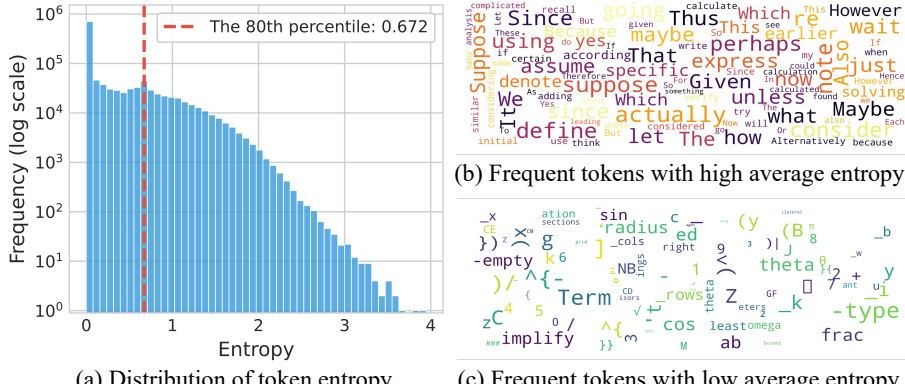

(a) Distribution of token entropy

(b) Frequent tokens with high average entropy

(c) Frequent tokens with low average entropy

Figure 2: Entropy patterns in the chain of thoughts of LLMs. **(a) Token entropy distribution.** The Y-axis frequency is on a *log scale*. A minority of tokens exhibit high entropy, while the majority have low entropy, often approaching zero. **(b) & (c) Word clouds of the top 100 tokens with the highest and lowest average entropy, respectively, selected from the set of frequently occurring tokens.** A larger font size indicates a higher average token entropy. Tokens with the highest average entropy typically function as "forks" to determine reasoning directions, whereas tokens with the lowest average entropy tend to execute reasoning steps along the established path.

**Entropy Pattern 1 in CoTs: Typically, only a minority of tokens are generated with high entropy, while a majority of tokens are outputted with low entropy.** We can observe in Figure 2(a) that the entropy of a large amount of tokens are quite small, and only a small amount of tokens have high entropy. Specifically, the entropy of over half the tokens (approximately 50.64%) is below $10^{-2}$, while only 20% of tokens have entropy greater than 0.672.

**Entropy Pattern 2 in CoTs: Tokens with the highest entropy typically serve to bridge the logical connection between two consecutive parts of reasoning, while tokens with the lowest entropy tend to complete the current part of a sentence or finish constructing a word. Other tokens combine these two functions to varying degrees.** In Figure 2(b) and (c), we select the 100 tokens generated with the highest average entropy and the lowest average entropy from a total of $10^6$ tokens, respectively. To mitigate the impact of noise on the average entropy, we only consider tokens with frequencies above 100. High-entropy tokens often act as logical connectors within and across sentences, such as "wait," "however," and "unless" (indicating contrasts or shifts), "thus" and "also" (showing progression or addition), or "since" and "because" (expressing causality). Similarly, tokens like "suppose," "assume," "given," and "define" frequently appear in mathematical derivations to introduce assumptions, known conditions, or definitions. Conversely, low-entropy tokens are often word suffixes, source code fragments, or mathematical expression components, all of which exhibit high determinism. Additionally, Figures 12 to 17 provide detailed visualization of token entropy in a long CoT, showing that most tokens outside the highest-entropy or lowest-entropy groups blend bridging and continuation functions to varying degrees.

**High-entropy tokens as "forks" in chain-of-thoughts** Based on the two observed patterns above, we refer to high-entropy tokens as *"forking tokens"*, as they often lead to different potential branches with high uncertainty in the reasoning process. To further confirm the role of forking tokens in a quantitative way, using Qwen3-8B [41] with a maximum response length of 28,672, we assign different decoding temperatures to the forking tokens and the other tokens in the evaluation on AIME 2024 and AIME 2025. Specifically, to analyze the effects of varying combinations of these temperatures on their behavior, we adjust probability distribution $\boldsymbol{p}'_t \in \mathbb{R}^V$ for each token $t$ as follows:

$$\boldsymbol{p}'_t = \text{Softmax}\left(\frac{\boldsymbol{z}_t}{T'_t}\right), \qquad \text{where} \quad T'_t = \begin{cases} T_{\text{high}} & \text{if } H_t > h_{\text{threshold}}, \\ T_{\text{low}} & \text{otherwise.} \end{cases} \tag{1}$$

Here, $\boldsymbol{z}_t \in \mathbb{R}^V$ denotes the pre-softmax logits for token $t$, and $T'_t \in \mathbb{R}$ represents the adjusted temperature for token $t$; $h_{\text{threshold}} = 0.672$ is the entropy threshold used to distinguish forking tokens from the other tokens, and is estimated by calculating the 80th percentile among the sampled $10^6$ tokens above; $T_{\text{high}} \in \mathbb{R}$ and $T_{\text{low}} \in \mathbb{R}$ correspond to the temperatures for forking tokens and the other tokens, respectively.

The effects of varying $T_{\text{high}}$ and $T_{\text{low}}$ are presented in Figure 3. It can be seen that lowering $T_{\text{high}}$ significantly degrades performance compared to lowering $T_{\text{low}}$. In contrast, increasing $T_{\text{high}}$ results in substantially better performance than increasing $T_{\text{low}}$, which can even cause LLMs generating nonsensical outputs. These results suggest that forking tokens benefit from being assigned a relatively higher temperature compared to other tokens. Given that forking tokens naturally exhibit higher entropy than other tokens, this further supports the need for them to operate at an even higher entropy level. This observation aligns with their role as "forks," where high entropy enables them to branch into diverse reasoning directions.

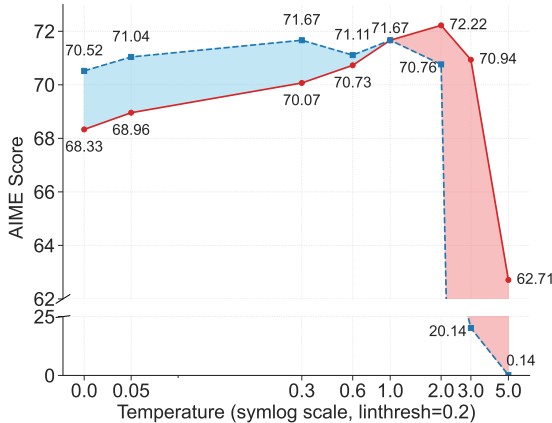

Figure 3: Average scores of AIME 2024 and AIME 2025. Red curve: varying $T_{\text{high}}$ with $T_{\text{low}} = 1$. Blue curve: varying $T_{\text{low}}$ with $T_{\text{high}} = 1$.

## 3 RLVR Preserves and Reinforces Base Model Entropy Patterns

In this section, building on the observations of entropy patterns in CoTs discussed in Section 2, we further investigate how these patterns evolve throughout RLVR training.

**RLVR primarily preserves the existing entropy patterns of the base models**  To analyze the evolution of entropy patterns during RLVR training, we apply DAPO [42] to the Qwen3-14B base model (details in Section 4). Using the reasoning model after RLVR, we generate 16 responses per question across the six benchmarks in Table 2. For each token in these responses, we compute logits using reasoning models from various RLVR stages and identify those in the top 20% entropy. We then calculate the overlap ratio (i.e., the fraction of shared top 20% high-entropy positions) between each intermediate model and both the base and final RLVR models. As shown in Table 1, although overlap with the base model gradually decreases and overlap with the final RLVR model increases, the base model's overlap still remains above 86% at convergence (step 1360), suggesting that RLVR largely retains the base models entropy patterns regarding which tokens exhibit high or low uncertainty.

Table 1: The progression of the overlap ratio in the positions of the top 20% high-entropy tokens, comparing the base model (i.e., step 0) with the model after RLVR training (i.e., step 1360).

| Compared w/ | Step 0 | Step 16 | Step 112 | Step 160 | Step 480 | Step 800 | Step 864 | Step 840 | Step 1280 | Step 1360 |
|---|---|---|---|---|---|---|---|---|---|---|
| Base Model | 100% | 98.92% | 98.70% | 93.04% | 93.02% | 93.03% | 87.45% | 87.22% | 87.09% | 86.67% |
| RLVR Model | 86.67% | 86.71% | 86.83% | 90.64% | 90.65% | 90.64% | 96.61% | 97.07% | 97.34% | 100% |

**RLVR predominantly alters the entropy of high-entropy tokens, whereas the entropy of low-entropy tokens remains comparatively stable with minimal variations.**  Using the same setup as Table 1, we compute the average entropy change after RLVR for each 5% entropy percentile range of the base model. It is observed in Figure 4 that tokens with higher initial entropy in the base model tend to exhibit larger increases in entropy after RLVR. This observation could also further reinforce that RLVR primarily preserves the entropy patterns of the base model. More experimental results and analyses are presented in Appendix C.1.

## 4 High-Entropy Minority Tokens Drive Effective RLVR

Reinforcement learning with verifiable rewards (RLVR) has become one of the most widely used approaches for training reasoning models [41, 6, 42, 44]. However, there is a lack of research on which types of tokens contribute the most to the learning of reasoning models. As highlighted in Section 2 and Section 3, high-entropy minority tokens are particularly important. In this section, we

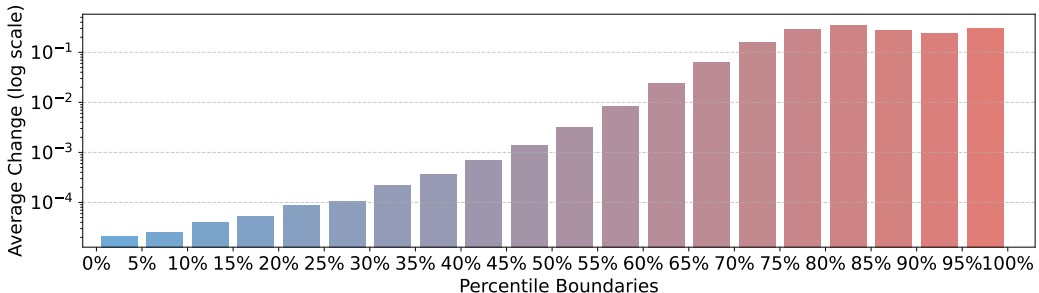

Figure 4: Average entropy change after RLVR within each 5% entropy percentile range of the base model. $x\%$ percentile means that $x\%$ of the tokens in the dataset have entropy values less than or equal to this value. It is worth noting that the Y-axis is presented on a *log scale*. Tokens with higher initial entropy tend to experience greater entropy increases after RLVR.

investigate the contribution of these high-entropy minority tokens, also referred to as forking tokens, on the development of reasoning capabilities during RLVR.

## 4.1 Formulation of RLVR Using Only Policy Gradients of the Highest-Entropy Tokens

Building on DAPO's objective in Equation (6), we discard the policy gradients of low-entropy tokens and train the model using only the policy gradients of high-entropy tokens. For each batch $\mathcal{B}$ sampled from the dataset $\mathcal{D}$, we calculate the maximum objective as:

$$\mathcal{J}^{\mathcal{B}}_{\text{HighEnt}}(\theta) = \mathbb{E}_{\mathcal{B}\sim\mathcal{D},(\boldsymbol{q},\boldsymbol{a})\sim\mathcal{B},\{\boldsymbol{o}^i\}^G_{i=1}\sim\pi_{\theta_{\text{old}}}(\cdot|\boldsymbol{q})}\left[\frac{1}{\sum^G_{i=1}\sum^{|\boldsymbol{o}^i|}_{t=1}\mathbb{I}\left[H^i_t\geq\tau^{\mathcal{B}}_\rho\right]}\sum^G_{i=1}\sum^{|\boldsymbol{o}^i|}_{t=1}\mathbb{I}\left[H^i_t\geq\tau^{\mathcal{B}}_\rho\right]\right.$$

$$\left.\cdot\min\left(r^i_t(\theta)\hat{A}^i_t,\,\text{clip}\left(r^i_t(\theta),1-\epsilon_{\text{low}},1+\epsilon_{\text{high}}\right)\hat{A}^i_t\right)\right],\,\text{s.t. }0<\left|\left\{\boldsymbol{o}^i\mid\texttt{is\_equivalent}(\boldsymbol{a},\boldsymbol{o}^i)\right\}\right|<G. \tag{2}$$

Here, $H^i_t$ denotes the entropy of token $t$ in response $i$, $\mathbb{I}[\cdot]$ is the indicator function that evaluates to 1 if the condition inside holds and 0 otherwise, $\rho\in(0,1]$ is a predefined ratio specifying the top proportion of high-entropy tokens to be selected within a batch, and $\tau^{\mathcal{B}}_\rho$ is the corresponding entropy threshold within the batch $\mathcal{B}$ such that only tokens with $H^i_t\geq\tau^{\mathcal{B}}_\rho$, comprising the top-$\rho$ fraction of all tokens in the batch, are used to compute the gradient.

Comparing Equation (2) with Equation (6), there are only two differences, as highlighted in red in Equation (2): (i) The advantage term is multiplied by $\mathbb{I}\left[H^i_t\geq\tau^{\mathcal{B}}_\rho\right]$, ensuring that within each (micro-)batch $\mathcal{B}$, only tokens $o^i_t$ whose corresponding entropy $H^i_t\geq\tau^{\mathcal{B}}_\rho$ are involved in the policy gradient loss calculation; (ii) The normalization term for the number of tokens is adjusted to $\sum^G_{i=1}\sum^{|\boldsymbol{o}^i|}_{t=1}\mathbb{I}\left[H^i_t\geq\tau^{\mathcal{B}}_\rho\right]$, meaning that only tokens whose entropy is at least the threshold $\tau^{\mathcal{B}}_\rho$ are considered.

## 4.2 Experimental Setup

**Training details** We adapt our training codebase from verl [32] and follow the training recipe of DAPO [42], one of the state-of-the-art RL algorithms for LLMs. Both configurations, RLVR with full gradients (vanilla DAPO depicted in Equation (6)) and RLVR with only policy gradients on forking tokens (described in Equation (2)), employ techniques such as clip-higher, dynamic sampling, token-level policy gradient loss, and overlong reward shaping [42]. For fair comparisons, we apply the same hyperparameters as recommended by DAPO: for clip-higher, $\epsilon_{\text{high}}=0.28,\epsilon_{\text{low}}=0.2$; for overlong reward shaping, the maximum response length is 20480 and the cache length is 4096. Furthermore, we use a training batch size of 512 and a mini-batch size of 32 in verl's configuration, resulting in 16 gradient steps per training batch, with a learning rate of $10^{-6}$ and no learning rate warmup or scheduling. Importantly, the training process excludes both KL divergence loss and entropy loss. To evaluate the scaling ability of these methods, we perform RLVR experiments across the Qwen3-32B base and Qwen3-8B base models, using DAPO-Math-17K [42] as the train-

Table 2: Comparison between *vanilla DAPO using all tokens* and *DAPO using only the top 20% high-entropy tokens (i.e. forking tokens)* in policy gradient loss, evaluated on the *Qwen3-32B*, *Qwen3-14B* and *Qwen3-8B* base models. "Acc@16" and "Len@16" denotes the average accuracy and response length over 16 evaluations per benchmark, respectively.

| Benchmark | DAPO w/ All Tokens | | DAPO w/ Forking Tokens | | Improvement | |
|---|---|---|---|---|---|---|
| | Acc@16 | Len@16 | Acc@16 | Len@16 | Acc@16 | Len@16 |
| **RLVR from the Qwen3-32B Base Model** | | | | | | |
| AIME'24 | 55.83 | 9644.15 | **63.54** | 12197.54 | **+7.71** | +2553.39 |
| AIME'25 | 45.63 | 9037.48 | **56.67** | 11842.25 | **+11.04** | +2804.77 |
| AMC'23 | 91.88 | 5285.03 | **94.22** | 5896.47 | **+2.34** | +611.44 |
| MATH500 | 94.36 | 2853.51 | **94.88** | 3366.01 | **+0.52** | +512.5 |
| Minerva | 45.70 | 2675.28 | **45.82** | 2759.88 | **+0.12** | +84.6 |
| Olympiad | 66.16 | 5597.37 | **69.02** | 7300.01 | **+2.86** | +1702.64 |
| **Average** | 66.59 | 5848.80 | **70.69** | 7227.03 | **+4.10** | +1378.22 |
| **RLVR from the Qwen3-14B Base Model** | | | | | | |
| AIME'24 | 45.21 | 7945.15 | **50.42** | 11814.36 | **+5.21** | +3869.21 |
| AIME'25 | 38.13 | 7056.98 | **42.92** | 12060.48 | **+4.79** | +5003.5 |
| AMC'23 | 89.53 | 4509.37 | **91.56** | 7095.13 | **+2.03** | +2585.76 |
| MATH500 | 92.23 | 2348.22 | **93.59** | 3970.10 | **+1.37** | +1621.88 |
| Minerva | 42.16 | 2011.16 | **43.20** | 2959.32 | **+1.03** | +948.16 |
| Olympiad | 61.14 | 4642.07 | **64.62** | 7871.25 | **+3.48** | +3229.18 |
| **Average** | 61.40 | 4752.16 | **64.39** | 7628.44 | **+2.99** | +2876.28 |
| **RLVR from the Qwen3-8B Base Model** | | | | | | |
| AIME'24 | 33.33 | 6884.89 | **34.58** | 9494.29 | **+1.25** | +2609.40 |
| AIME'25 | 25.42 | 5915.91 | **26.25** | 8120.20 | **+0.83** | +2204.29 |
| AMC'23 | **77.81** | 3967.91 | 77.19 | 5450.62 | -0.625 | +1482.71 |
| MATH500 | 89.24 | 2059.00 | **89.70** | 2672.91 | **+0.46** | +613.91 |
| Minerva | 39.77 | 1450.68 | **40.26** | 2068.41 | **+0.48** | +617.73 |
| Olympiad | 56.67 | 3853.55 | **57.43** | 5241.54 | **+0.76** | +1387.99 |
| **Average** | 53.71 | 4021.99 | **54.23** | 5508.00 | **+0.53** | +1486.01 |

ing dataset. For main results, we set $\rho = 20\%$ in Equation (2), meaning that the policy is updated using only the gradients of the top 20% highest-entropy tokens within each batch. The chat template we use for Qwen3 models is "User:\n[question]\nPlease reason step by step, and put your final answer within \boxed{}.\n\nAssistant:\n" with "<|endoftext|>" serving as the EOS token, where "[question]" should be replaced by the specific question.

**Evaluation** We evaluate our models on 6 standard mathematical reasoning benchmarks commonly used for assessing reasoning capabilities: AIME'24, AIME'25, AMC'23, MATH500 [12], Minerva, and OlympiadBench [11]. All evaluations are conducted in a zero-shot setting. For each question, we generate 16 independent responses under a decoding temperature $T = 1.0$, and report the average accuracy and the average number of tokens per response.

### 4.3 Main Results

We present the main experimental results below. Additional results are provided in Appendix C.

**High-entropy tokens drive reinforcement learning for LLM reasoning.** Figure 5 and Table 2 compare vanilla DAPO which uses all tokens, and our approach that retains only the top 20% high-entropy tokens in the policy gradient loss. Surprisingly, discarding the bottom 80% low-entropy tokens does not degrade reasoning performance and can even lead to improvements across six benchmarks. On the Qwen3-32B base model, this approach delivers gains of 7.71 points on AIME'24 and

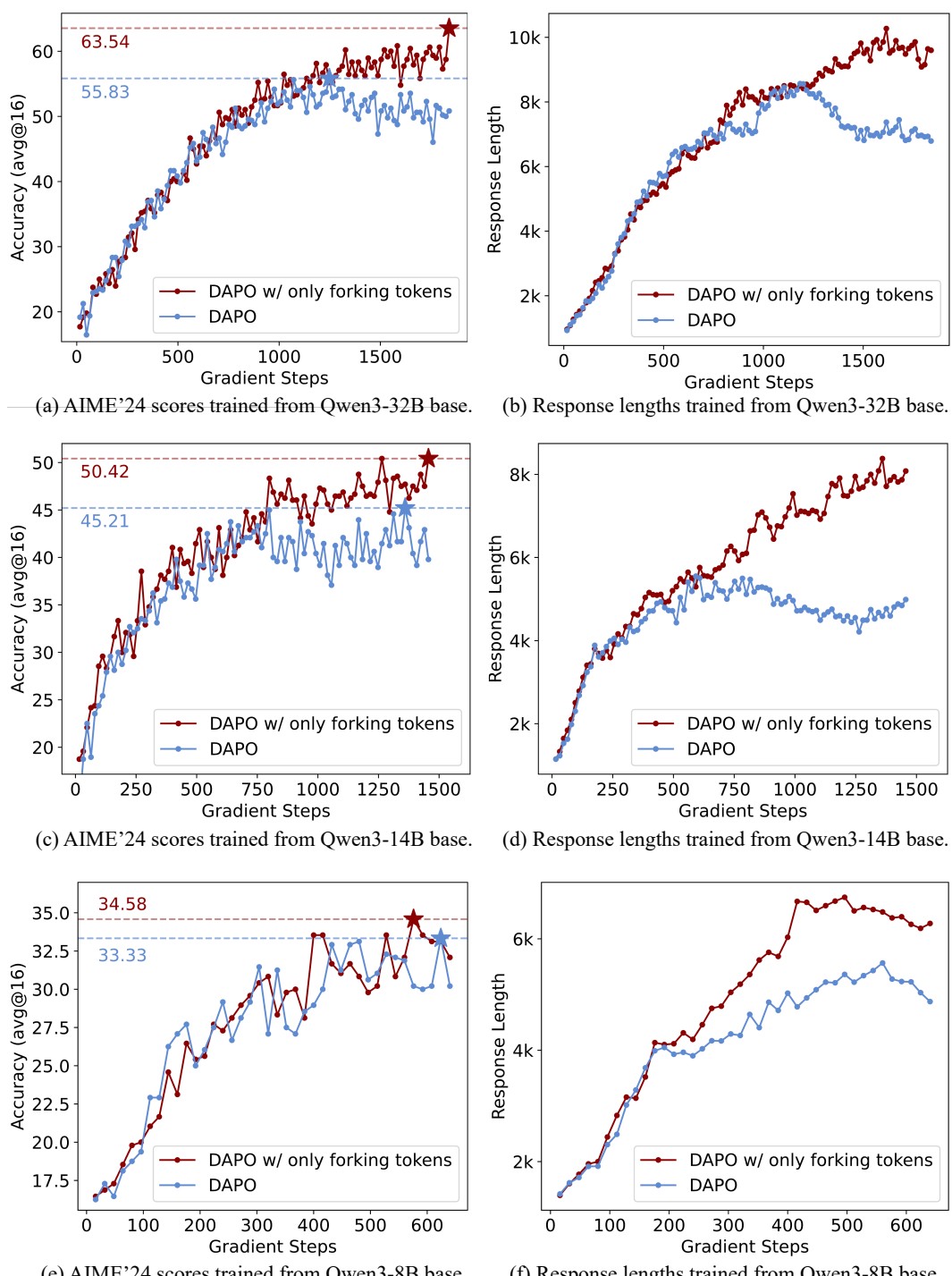

(a) AIME'24 scores trained from Qwen3-32B base.

(b) Response lengths trained from Qwen3-32B base.

(c) AIME'24 scores trained from Qwen3-14B base.

(d) Response lengths trained from Qwen3-14B base.

(e) AIME'24 scores trained from Qwen3-8B base.

(f) Response lengths trained from Qwen3-8B base.

Figure 5: A comparison of *vanilla DAPO with full tokens* and *DAPO with top 20% high-entropy (forking) tokens* in policy gradient loss was conducted on *Qwen3-32B*, *Qwen3-14B*, and *Qwen3-8B* models. **(a) & (b) Qwen3-32B:** Dropping the bottom 80% low-entropy tokens stabilizes training and improves the AIME'24 score by 7.73. **(c) & (d) Qwen3-14B:** Similarly, removing 80% low-entropy tokens yields a 5.21 increase in the AIME'24 score. **(e) & (f) Qwen3-8B:** Retaining only the top 20% forking tokens maintains performance. Additionally, using only the top 20% high-entropy tokens increases response length across all model sizes.

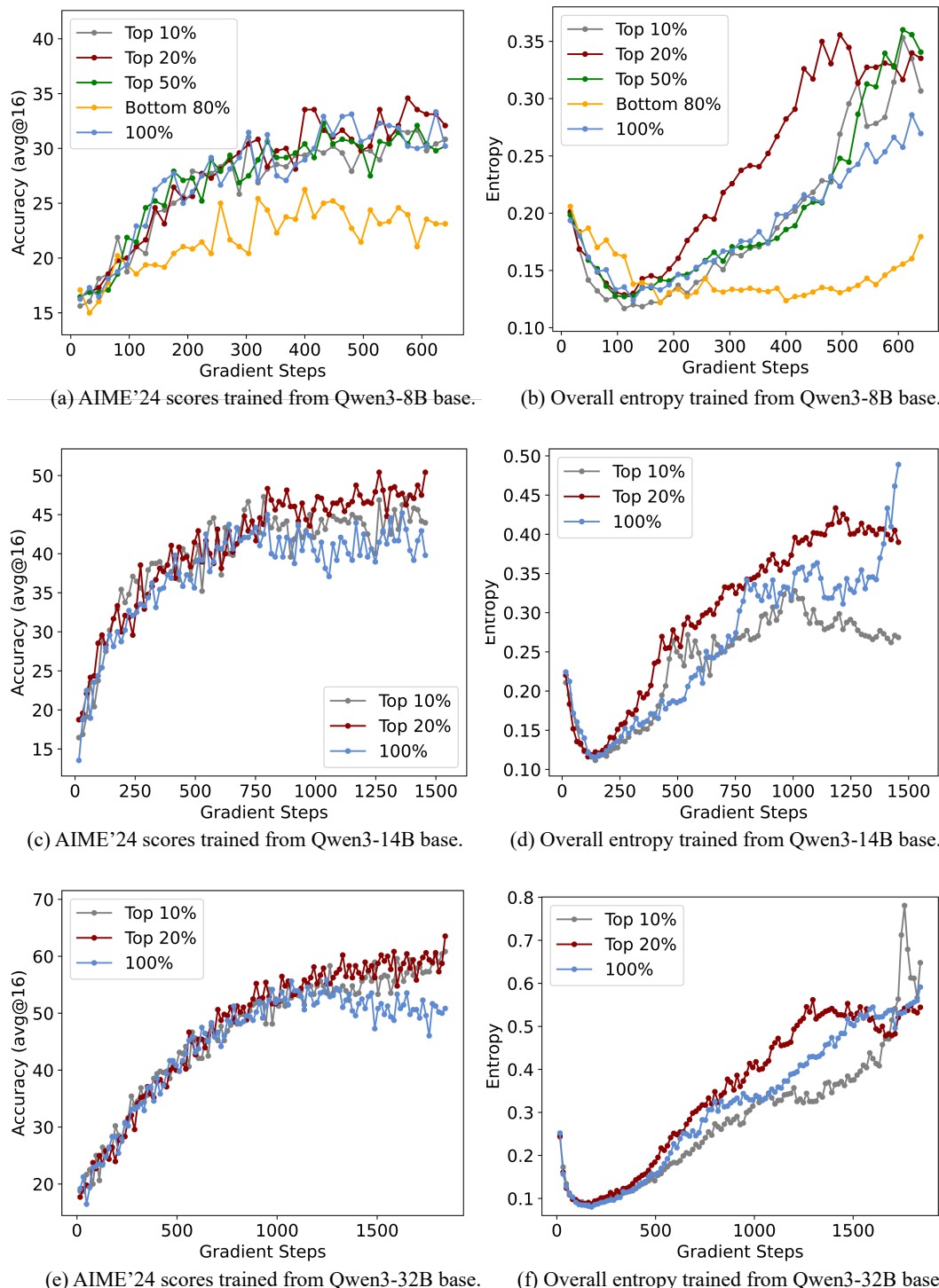

(a) AIME'24 scores trained from Qwen3-8B base.

(b) Overall entropy trained from Qwen3-8B base.

(c) AIME'24 scores trained from Qwen3-14B base.

(d) Overall entropy trained from Qwen3-14B base.

(e) AIME'24 scores trained from Qwen3-32B base.

(f) Overall entropy trained from Qwen3-32B base.

Figure 6: Comparison among DAPO using different range of tokens in policy gradient loss. Top $x\%$ means only using the $x\%$ of the tokens with highest entropy ($x = 10, 20, 50$), bottom $80\%$ means only using the $80\%$ of the tokens with lowest entropy, and $100\%$ means using all tokens (i.e., vanilla DAPO). Furthermore, "overall entropy" refers to the average entropy over all tokens.

11.04 points on AIME'25. Similarly, the Qwen3-14B base model shows improvements of 5.21 points on AIME'24 and 4.79 points on AIME'25. For the Qwen3-8B base model, performance remains unaffected. These findings suggest that the gains in reasoning ability during RLVR are driven

primarily by high-entropy tokens, while low-entropy tokens may have little effect on or could even hinder reasoning performance, particularly on the Qwen3-32B and Qwen3-14B base models.

To conduct a deeper analysis, we vary the proportion, denoted as $\rho$ in Equation (2), for experiments, as shown in Figure 6(a). The results show that the performance of the Qwen3-8B base model remains relatively consistent across different proportions, such as 10%, 20%, and 50%. For the Qwen3-14B and 32B base models, Figure 6(c) and (e) reveal that reducing $\rho$ from 20% to 10% leads to a slight drop in performance, while increasing it to 100% results in a notable decline. These observations indicate that within a reasonable range, reasoning performance is largely *insensitive* to the exact value of $\rho$. More importantly, they suggest that focusing on high-entropy tokens, rather than using all tokens, generally preserves performance, and could even offer substantial gains in larger models.

**Low-entropy tokens contribute minimally to reasoning performance.** As illustrated in Figure 6(a) and (b), retaining only the bottom 80% of tokens with low entropy during RLVR leads to a substantial decline in performance, even though these tokens account for 80% of the total token count used in training. This finding indicates that low-entropy tokens contribute minimally to enhancing reasoning capabilities, highlighting the greater importance of high-entropy tokens for effective model training.

**The effectiveness of high-entropy tokens may lie in their ability to enhance exploration.** Our analysis reveals that focusing on a subset of high-entropy tokens, approximately 20% in our experiments, strikes an effective balance between exploration and training stability in RLVR. As illustrated in Figure 6(b), (d) and (f), adjusting the ratio $\rho$ from 20% to either 10%, 50%, or 100% leads to persistently lower overall entropy starting from the early training phase and continuing up to the point where performance begins to converge. Moreover, training with the bottom 80% of low-entropy tokens results in significantly reduced overall entropy. These findings indicate that retaining a certain proportion of high-entropy tokens may facilitate effective exploration. Tokens outside this range could be less helpful or possibly even detrimental to exploration, particularly during the critical phase before performance convergence. This might explain why, on the Qwen3-32B base model, DAPO using only the top 20% high-entropy tokens outperforms vanilla DAPO, as shown in Figure 5(a). However, on the Qwen3-8B base model, probably due to the models lower capacity, the benefits of enhanced exploration appear limited.

## 5   Conclusion

In this work, we analyze RLVR through a novel perspective of token entropy, providing fresh insights into the mechanisms of reasoning in LLMs. Our study of CoT reasoning shows that only a small subset of tokens exhibit high entropy and serve as forks in reasoning paths that influence reasoning directions. Additionally, our analysis of entropy dynamics during RLVR training reveals that the reasoning model largely retains the base model's entropy patterns, with RLVR mainly modifying the entropy of already high-entropy tokens. Building on these findings, which underscore the significance of high-entropy minority tokens, we restrict policy gradient updates in RLVR to the top 20% highest-entropy tokens. This approach achieves performance comparable to, or even surpassing, full-token RLVR training, while exhibiting a strong scaling trend with model size. In contrast, directing optimization toward the low-entropy majority results in a significant decline in performance. These findings indicate that RLVR's effectiveness stems primarily from optimizing this high-entropy subset, suggesting more focused and efficient strategies for improving LLM reasoning capabilities.

**Limitations**   We believe there is still room for improvement in our work. First, our experiments could be extended to models beyond the Qwen family. Although we attempted to evaluate our approach on LLaMA models, they struggled to achieve meaningful performance on the AIME benchmarks. Additionally, the scope of our dataset could be expanded to encompass domains beyond mathematics, such as programming or more complex tasks like ARC-AGI [3, 4]. Furthermore, our findings are based on specific experimental settings, and it is possible that the observations and conclusions presented in this paper may not generalize to all RLVR scenarios. For instance, in a different RLVR setting, the effective proportion of 20% observed in our experiments may need to be adjusted to a different value to achieve optimal results. Future directions involve developing new RLVR algorithms to better leverage high-entropy minority tokens and exploring how these insights can enhance not only RLVR but also other approaches, such as supervised fine-tuning (SFT), distillation [13], inference, and multi-modal training [2, 20, 27].

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

# Appendices

## Contents

# A Preliminaries

## A.1 Token Entropy Calculation

The token-level generation entropy (referred to as *token entropy* for brevity) for token $t$ is defined as

$$H_t := -\sum_{j=1}^{V} p_{t,j} \log p_{t,j}, \quad \text{where} \quad (p_{t,1}, \cdots, p_{t,V}) = \boldsymbol{p}_t = \pi_{\boldsymbol{\theta}}(\cdot \mid \boldsymbol{q}, \boldsymbol{o}_{<t}) = \text{Softmax}\left(\frac{\boldsymbol{z}_t}{T}\right). \quad (3)$$

Here, $\pi_{\boldsymbol{\theta}}$ denotes an LLM parameterized by $\boldsymbol{\theta}$, $\boldsymbol{q}$ is the input query, and $\boldsymbol{o}_{<t} = (o_1, o_2, \cdots, o_{t-1})$ represents the previously generated tokens. $V$ is the vocabulary size, $\boldsymbol{z}_t \in \mathbb{R}^V$ denotes the pre-softmax logits at time step $t$, $\boldsymbol{p}_t \in \mathbb{R}^V$ is the corresponding probability distribution over the vocabulary, and $T \in \mathbb{R}$ is the decoding temperature. In off-policy settings, sequences are generated by a rollout policy $\pi_{\boldsymbol{\phi}}$ while the training policy is $\pi_{\boldsymbol{\theta}}$, with $\boldsymbol{\phi} \neq \boldsymbol{\theta}$. The entropy is still calculated using $\pi_{\boldsymbol{\theta}}$, as defined in Equation (3), to measure the uncertainty of the training policy in the given sequence.

**"Token entropy" corresponds to the token generation distribution, not a specific token.** Throughout our paper, we clarify that the token entropy $H_t$ refers to the entropy at index $t$, which is determined by the token generation distribution $\boldsymbol{p}_t$ rather than by any specific token $o_t$ sampled from $\boldsymbol{p}_t$. For brevity, when discussing the token $o_t$ sampled from the distribution $\boldsymbol{p}_t$, we describe its associated entropy as $H_t$ and refer to $H_t$ as the token entropy of $o_t$. However, if there exists another index $t' \neq t$ such that $o_{t'} = o_t$, the token entropy of $o_{t'}$ is not necessarily equal to $H_t$.

## A.2 RLVR Algorithms

**Proximal Policy Optimization (PPO)** PPO [30] is a widely adopted policy gradient algorithm in RLVR. To stabilize training, PPO restricts policy updates to remain within a proximal region of the old policy $\pi_{\boldsymbol{\theta}_{\text{old}}}$ using the following clipped surrogate to maximize the objective:

$$J_{\text{PPO}}(\theta) = \mathbb{E}_{\mathcal{B} \sim \mathcal{D}, (\boldsymbol{q}, \boldsymbol{a}) \sim \mathcal{B}, \boldsymbol{o} \sim \pi_{\boldsymbol{\theta}_{\text{old}}}(\cdot|\boldsymbol{q})} \left[ \min\left( r_t(\theta)\hat{A}_t, \text{clip}(r_t(\theta), 1 - \epsilon, 1 + \epsilon)\hat{A}_t \right) \right],$$

$$\text{with} \quad r_t(\boldsymbol{\theta}) = \frac{\pi_{\boldsymbol{\theta}}(o_t|\boldsymbol{q}, \boldsymbol{o}_{<t})}{\pi_{\boldsymbol{\theta}_{\text{old}}}(o_t|\boldsymbol{q}, \boldsymbol{o}_{<t})}. \quad (4)$$

Here, $\mathcal{D}$ is a dataset of queries $\boldsymbol{q}$ and corresponding ground-truth answers $\boldsymbol{a}$, $\mathcal{B}$ is a batch sampled from $\mathcal{D}$, $\epsilon \in \mathbb{R}$ is a hyperparameter typically set to $0.2$, and $\hat{A}_t$ is the estimated advantage computed using a value network.

**Group Relative Policy Optimization (GRPO)** Building on the clipped objective in Equation (4), GRPO [31] discards the value network by estimating advantages using the average reward within a group of sampled responses. Specifically, for each query $\boldsymbol{q}$ and its ground-truth answer $\boldsymbol{a}$, the rollout policy $\pi_{\boldsymbol{\theta}\text{old}}$ generates a group of responses $\{\boldsymbol{o}^i\}_{i=1}^{G}$ with corresponding outcome rewards $\{R^i\}_{i=1}^{G}$, where $G \in \mathbb{R}$ is the group size. The estimated advantage $\hat{A}_t^i$ is then computed as:

$$\hat{A}_t^i = \frac{r^i - \text{mean}(\{R^i\}_{i=1}^{G})}{\text{std}(\{R^i\}_{i=1}^{G})}, \quad \text{where} \quad R^i = \begin{cases} 1.0 & \text{if is\_equivalent}(\boldsymbol{a}, \boldsymbol{o}^i), \\ 0.0 & \text{otherwise.} \end{cases} \quad (5)$$

In addition to this modified advantage estimation, GRPO adds a KL penalty term to the clipped objective in Equation (4).

**Dynamic sAmpling Policy Optimization (DAPO)** Building on GRPO, DAPO [42] removes the KL penalty, introduces a clip-higher mechanism, incorporates dynamic sampling, applies a token-level policy gradient loss, and adopts overlong reward shaping, leading to the following maximization objective, where $r_t^i(\boldsymbol{\theta})$ is defined as in Equation (4), and $\hat{A}_t^i$ is computed as in Equation (5):

$$\mathcal{J}_{\text{DAPO}}(\theta) = \mathbb{E}_{\mathcal{B} \sim \mathcal{D}, (\boldsymbol{q}, \boldsymbol{a}) \sim \mathcal{B}, \{\boldsymbol{o}^i\}_{i=1}^{G} \sim \pi_{\boldsymbol{\theta}_{\text{old}}}(\cdot|\boldsymbol{q})} \left[ \frac{1}{\sum_{i=1}^{G} |\boldsymbol{o}^i|} \sum_{i=1}^{G} \sum_{t=1}^{|\boldsymbol{o}^i|} \min\left( r_t^i(\theta)\hat{A}_t^i, \right. \right.$$

$$\left. \left. \text{clip}\left(r_t^i(\theta), 1 - \epsilon_{\text{low}}, 1 + \epsilon_{\text{high}}\right)\hat{A}_t^i \right) \right], \quad \text{s.t.} \quad 0 < \left|\left\{\boldsymbol{o}^i \mid \text{is\_equivalent}(\boldsymbol{a}, \boldsymbol{o}^i)\right\}\right| < G. \quad (6)$$

DAPO is one of the state-of-the-art RLVR algorithms without a value network. In this work, we use DAPO as the baseline for our RLVR experiments.

## B    Related Work

**Reinforcement Learning for LLM**    Before the advent of reasoning-capable models like OpenAI's o1 [24], reinforcement learning (RL) was widely used in reinforcement learning from human feedback (RLHF) to improve large language models' (LLMs) instruction-following and alignment with human preferences [25]. RLHF methods are broadly categorized into online and offline preference optimization. Online methods, such as PPO [30], GRPO [31], and REINFORCE [39], generate responses during training and receive real-time feedback. Offline methods like DPO [28], SimPO [22], and KTO [7] optimize policies using pre-collected preferences, typically from human annotators or LLMs. While offline methods are more training-efficient, they often underperform compared to online approaches [33]. Recently, RL with verifiable rewards (RLVR)[17] has emerged as a promising approach for enhancing reasoning in LLMs, particularly in domains like mathematics and programming[31, 6, 41, 17]. OpenAI o1 [24] was the first to show that RL can effectively incentivize reasoning at scale. Building on o1, models such as DeepSeek R1 [6], QwQ [35], Kimi k1.5 [34], and Qwen3 [41] have aimed to match or exceed its performance. DeepSeek R1 stands out for showing that strong reasoning can emerge through outcome-based optimization using the online RL algorithm GRPO [31]. It also introduced a zero RL paradigm, where reasoning abilities are elicited from the base model without conventional RL fine-tuning. Inspired by these results, subsequent methods such as DAPO [42], VAPO [44], SimpleRLZoo [45], and Open-Reasoner-Zero [15] have further explored RL-based reasoning. In this work, we use DAPO as our baseline to investigate key aspects of RL applied to LLMs.

**Analysis on Reinforcement Learning with Verifiable Rewards**    Recently, Reinforcement Learning with Verifiable Rewards (RLVR) has emerged as a prevalent approach to enhance the reasoning capabilities of large language models (LLMs). Several studies have analyzed the characteristics of RLVR and its related concepts. Gandhi et al. [8] find that the presence of reasoning behaviors, rather than the correctness of answers, is the key factor driving performance improvements in reinforcement learning. Similarly, Li et al. [18] show that the structure of long chains of thought (CoT) is critical to the learning process, while the content of individual reasoning steps has minimal impact. Vassoyan et al. [36] identify "critical tokens" in CoTs, which are decision points where models are prone to errors, and propose encouraging exploration around these tokens by modifying the KL penalty. Lin et al. [19] also identify critical tokens that significantly influence incorrect outcomes and demonstrate that identifying and replacing these tokens can alter model behavior. Our finding that RLVR primarily focuses on forking tokens in reasoning paths may share some common ground with the observations of Gandhi et al. [8] and Li et al. [18], who suggest that RLVR primarily learns the format rather than the content. However, our analysis goes further by identifying the finding at the token level. Moreover, the concept of critical tokens in Vassoyan et al. [36] and Lin et al. [19] is closely related to the high-entropy minority tokens we introduce. In contrast to prior work, which judges token importance based on correctness of the output, we propose token entropy as a criterion that may more accurately reflect the underlying mechanisms of LLMs.

## C    More Results

In this part, we include some additional results as a supplement to Sections 2 to 4.

### C.1    Additional Results to Figure 4

Additional to Figure 4, Figure 7 further illustrates the evolution of entropy percentiles during RLVR training using the Qwen3-14B base model. The figure reveals that as we move from the 0th to the 100th percentile, the range of fluctuations during the whole RLVR training steadily diminishes. These observations suggest that throughout the whole training process, RLVR primarily adjusts the entropy of high-entropy tokens, while the entropy of low-entropy tokens exhibits minor variation and remains relatively stable.

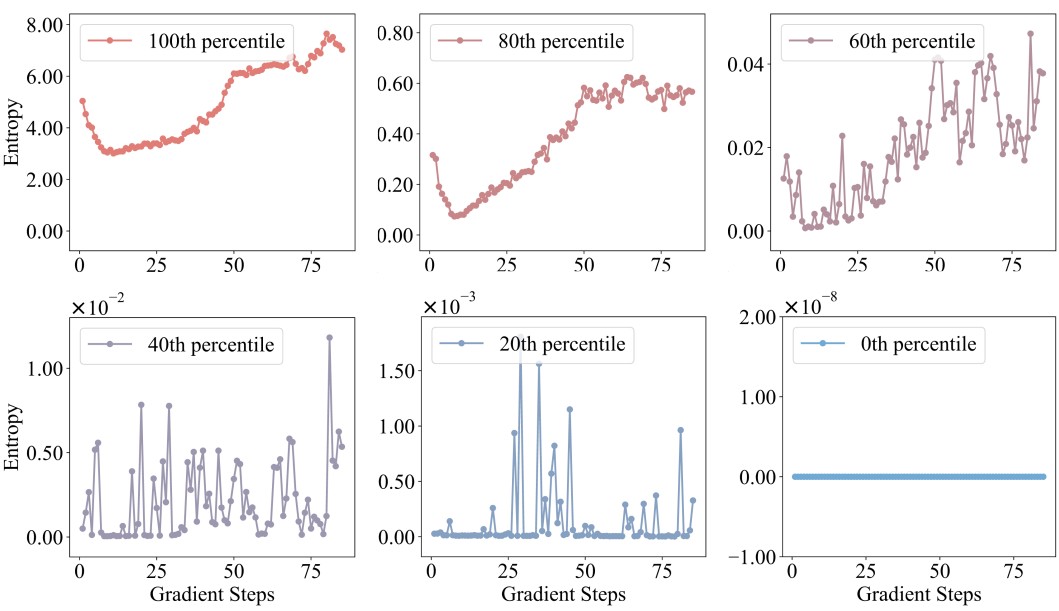

Figure 7: The evolution of entropy percentiles during RLVR training. $x$-th percentile means that $x\%$ of the tokens in the dataset have entropy values less than or equal to this value. In other words, it represents the threshold below which the entropy values of the lowest $x\%$ of tokens fall, allowing us to track how different segments of the entropy distribution change throughout training.

## C.2  Scaling Trend of Retaining Only Forking Tokens

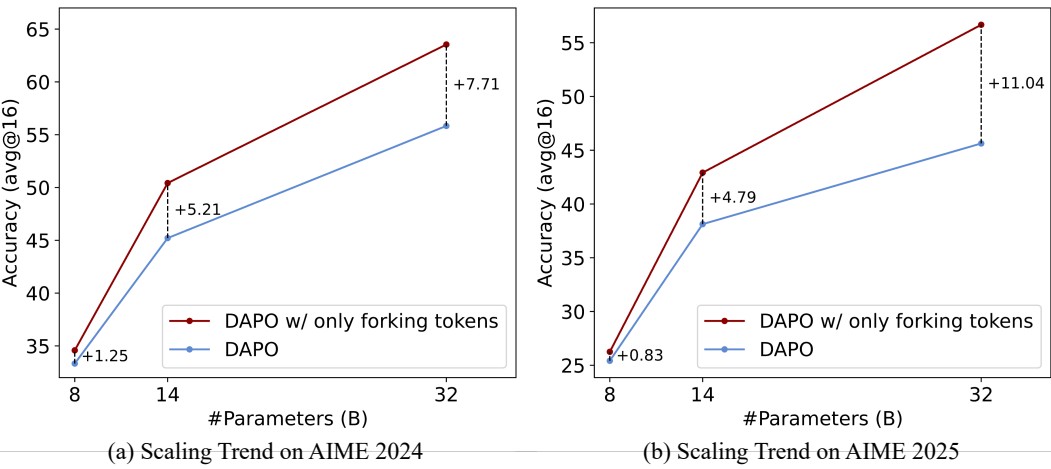

(a) Scaling Trend on AIME 2024         (b) Scaling Trend on AIME 2025

Figure 8: Scaling trend of DAPO using only forking tokens (i.e., top 20% of high-entropy tokens) in policy gradient loss.

In this part, we present the scaling trend when utilizing only forking tokens, as illustrated in Figure 8. On the AIME'24 and AIME'25 benchmarks, we observe that as the model size increases, the performance gain over vanilla DAPO becomes increasingly significant. This suggests a promising conclusion: **focusing solely on forking tokens in the policy gradient loss could offer greater advantages in larger reasoning models.**

## C.3  Generalization Ability to Other Domains

As outlined in Section 4.2, we used the DAPO-Math-17K dataset, which primarily consists of mathematical data, for our RLVR experiments. Here, we test whether DAPO, when trained on a math

dataset and using only a small fraction of high-entropy tokens in the policy gradient loss, can still surpass vanilla DAPO on out-of-distribution test sets, such as LiveCodeBench [16].

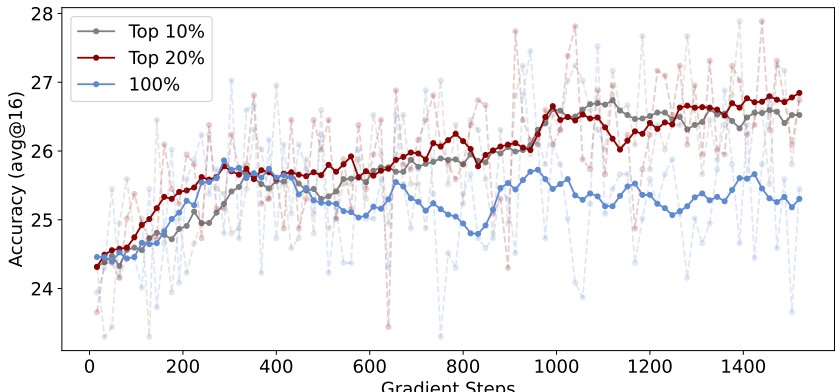

Figure 9: Comparison among DAPO using different range of tokens in policy gradient loss trained from the Qwen3-32B base model **on the out-of-distribution LiveCodeBench Benchmark** [16] (v5, Aug. 2024 to Feb. 2025). Top $x\%$ means only using the $x\%$ of the tokens with highest entropy ($x = 10, 20$), and $100\%$ means using all tokens (i.e. vanilla DAPO). Due to the high variance of the accuracy curves, we smooth the curves using window smoothing with a window size of 10.

The results comparing DAPO with only top $10\%$ or $20\%$ tokens with highest entropy to vanilla DAPO (which uses $100\%$ tokens) on the Qwen3-32B base are illustrated in Figure 9, using the same setup described in Section 4.2. From these results, we observe that even when retaining only top $10\%$ or $20\%$ tokens with highest entropy, DAPO still significantly outperforms vanilla DAPO on the out-of-distribution test dataset LiveCodeBench. **This finding suggests that high-entropy tokens may be associated with the generalization capabilities of reasoning models. Retaining only a small subset of tokens with the highest entropy could potentially enhance the generalization ability of reasoning models.**

### C.4   Unlocking more potential of RLVR with only forking tokens

In the experiments described in Section 4.3, we set a maximum response length of 20480. As shown in Figure 10, we increased the maximum response length for DAPO w/ only forking tokens (depicted in Figure 5(a) and (b))trained from the Qwen3-32B base modelto 29696. This adjustment resulted in an improvement of the already SoTA performance on AIME'24, increasing from 63.54 to 68.12. These findings suggest that the full potential of our approach may not yet be realized, and with a longer context length or potentially more challenging training data, even greater performance gains could be achieved.

### C.5   Results on models other than Qwen

We compare DAPO using only forking tokens (i.e., the top 20% tokens with the highest entropy) against vanilla DAPO on models other than the Qwen series, specifically the Llama-3.1-8B model. When DAPO is applied to the Llama-3.1-8B base model, we observe that it achieves very low accuracy (approximately 1%) on the training dataset (i.e., DAPO-MATH-17K [42]) and often generates responses with repetitive words early in the RL training process. To address this, we use the Qwen3-32B model [41] as a teacher to generate responses for DAPO-MATH-17K queries. From the generated queries, we randomly sample 10,000 with correct answers to serve as cold-start data and perform supervised fine-tuning (SFT) on the Llama-3.1-8B base model [9]. The remaining 7,398 queries are reserved for RL after the cold-start phase. The AIME'24 score and response length during RL training are plotted in Figure 11. The results indicate that DAPO with only forking tokens still surpasses vanilla DAPO, while also producing longer responses on average. However, given the relatively low performance of both configurations on AIME'24, we believe the results on Llama-3.1-8B are less convincing compared to those observed on the Qwen3 models.

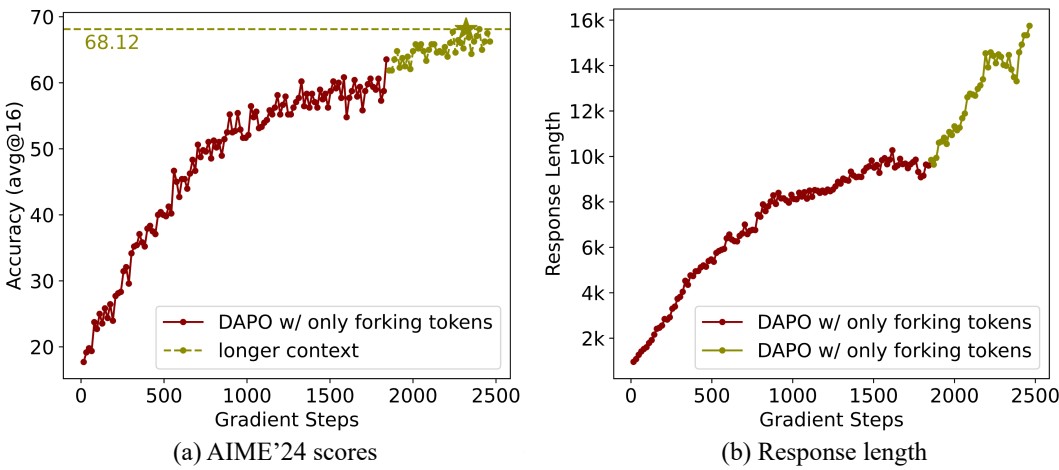

(a) AIME'24 scores

(b) Response length

Figure 10: By extending the maximum response length from 20,480 to 29,696 and continuing training from the SoTA 32B model shown in Figure 5(a) and (b), the AIME'24 scores improve further from 63.54 to 68.12, alongside a notable increase in response length.

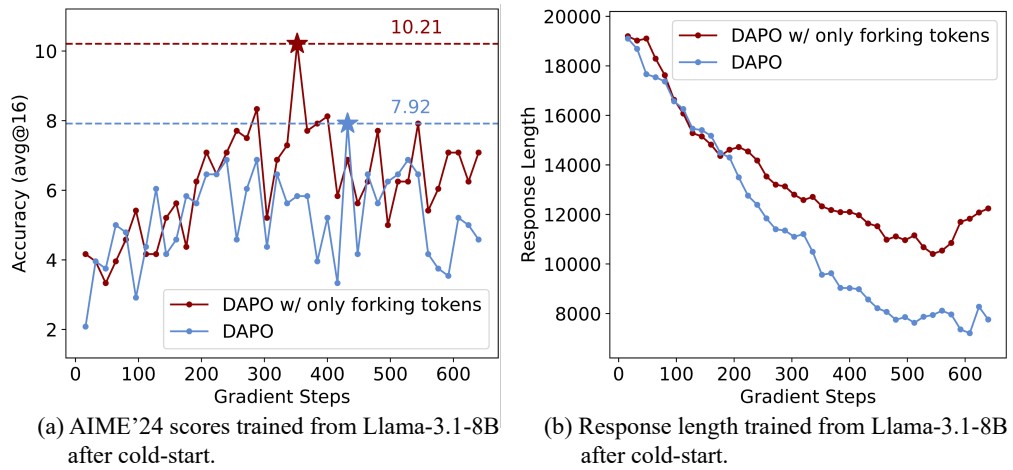

(a) AIME'24 scores trained from Llama-3.1-8B after cold-start.

(b) Response length trained from Llama-3.1-8B after cold-start.

Figure 11: Comparison of DAPO using only forking tokens and vanilla DAPO, both trained from Llama-3.1-8B after cold-start.

### C.6 Qualitative Results

**Visualization of token entropy of a full CoT response**    We visualize the token entropy of a long, randomly generated Chain of Thought (CoT) response by Qwen3-8B [41] in Figure 12 to Figure 17. From the visualization, we observe that high-entropy tokens are relatively sparse among all tokens. These high-entropy tokens often act as branching points in reasoning paths, influencing the direction of reasoning. This observation further complements the entropy patterns discussed in Section 2.

## D    Discussions

**Discussion 1: High-entropy minority tokens (i.e., forking tokens) could play a key role in explaining why RL generalizes while SFT memorizes.**    [5] demonstrated empirically that RL, particularly with outcome-based rewards, exhibits strong generalization to unseen, rule-based tasks, whereas supervised fine-tuning (SFT) is prone to memorizing training data and struggles with generalization outside the training distribution. We hypothesize that one critical factor underlying the differing generalization capabilities of RL and SFT may be related to entropy in forking tokens. Our experiments (e.g., Figure 7 and Figure 6) suggest that RL tends to preserve or even increase the

entropy of forking tokens, maintaining the flexibility of reasoning paths. In contrast, SFT pushes output logits towards one-hot distributions, leading to reduced entropy in forking tokens and, consequently, a loss of reasoning path flexibility. This flexibility may be a crucial determinant of a reasoning model's ability to generalize effectively to unseen tasks.

**Discussion 2: Unlike traditional RL, LLM reasoning integrates prior knowledge and must produce readable output. Consequently, LLM CoTs contain a mix of low-entropy majority tokens and high-entropy minority tokens, whereas traditional RL can assume uniform action entropy throughout a trajectory.** As shown in Figure 2(a), most LLM CoT tokens have low entropy, with only a small fraction exhibiting high entropy. In contrast, traditional RL typically formulates each action distribution as Gaussian with a predefined standard deviation [30, 29, 38], resulting in uniform entropy across actions. We attribute this distinct entropy pattern in LLM CoTs to their pretraining on large-scale prior knowledge and the need for language fluency. This forces most tokens to align with memorized linguistic structures, yielding low entropy. Only a small set of tokens that are inherently uncertain in the pretraining corpus allows for exploration, and thus exhibits high entropy. This deduction is consistent with our results in Table 1.

**Discussion 3: In RLVR, entropy bonus may be suboptimal, as it increases the entropy of low-entropy majority tokens. In contrast, clip-higher effectively promotes entropy in high-entropy minority tokens.** In RL, entropy bonus is commonly added to the training loss to encourage exploration by increasing the entropy of actions—a well-established practice in traditional tasks [30, 39, 23], and recently applied to LLM reasoning [32, 14]. However, as discussed above, unlike typical RL trajectories, LLM CoTs display distinct entropy patterns. Increasing entropy across all tokens can degrade performance by disrupting the low-entropy majority, while selectively increasing the entropy of high-entropy minority tokens improves performance (Figure 3). Thus, uniformly applied entropy bonuses are suboptimal for CoT reasoning. Instead, clip-higher [42], which moderately raises $\epsilon_{\text{high}}$ in Equation (6), better targets high-entropy tokens. Empirically, we observe that tokens with high importance ratios $r_t(\theta)$ (as defined in Equation (4)) tend to have higher entropy. By including more of these tokens in training, clip-higher increases overall entropy without significantly affecting low-entropy tokens, as supported by [42] and illustrated in Figure 7.

# E    Compute Resources

We utilize NVIDIA A100 80G GPUs as our compute resources for conducting experiments. Each experiment involving 8B, 14B, and 32B models is executed on 8 nodes, with each node equipped with 8 GPUs. Our training codebase is adapted from verl [32]. For the 8B model, the initial processing time for each mini-batch is approximately 0.5 hours, which increases to around 1.5 hours as the response length expands during training. Similarly, for the 14B model, the runtime per mini-batch starts at about 0.8 hours and grows to approximately 2.2 hours as the response length increases. In the case of the 32B model, processing each mini-batch initially takes about 1 hour, but this extends to roughly 3 hours as the response length continues to grow during training.

# F    Broader Impacts and LLM Usage Disclosure

**Broader impacts**   Our study analyzes entropy patterns in CoTs and enhances LLM reasoning by retaining only the top 20% of high-entropy tokens in policy gradient loss during RLVR training. While this improves reasoning abilities, it may also increase the risk of hallucinations or unsafe outputs. We urge developers to responsibly apply our findings to build safe reasoning models and advise users to exercise caution when using them.

**Declaration of LLM Usage**   LLMs are the primary subject of research in this paper, and all of our experiments are conducted on LLMs. Additionally, we also used LLMs for polishing the text during the writing process.

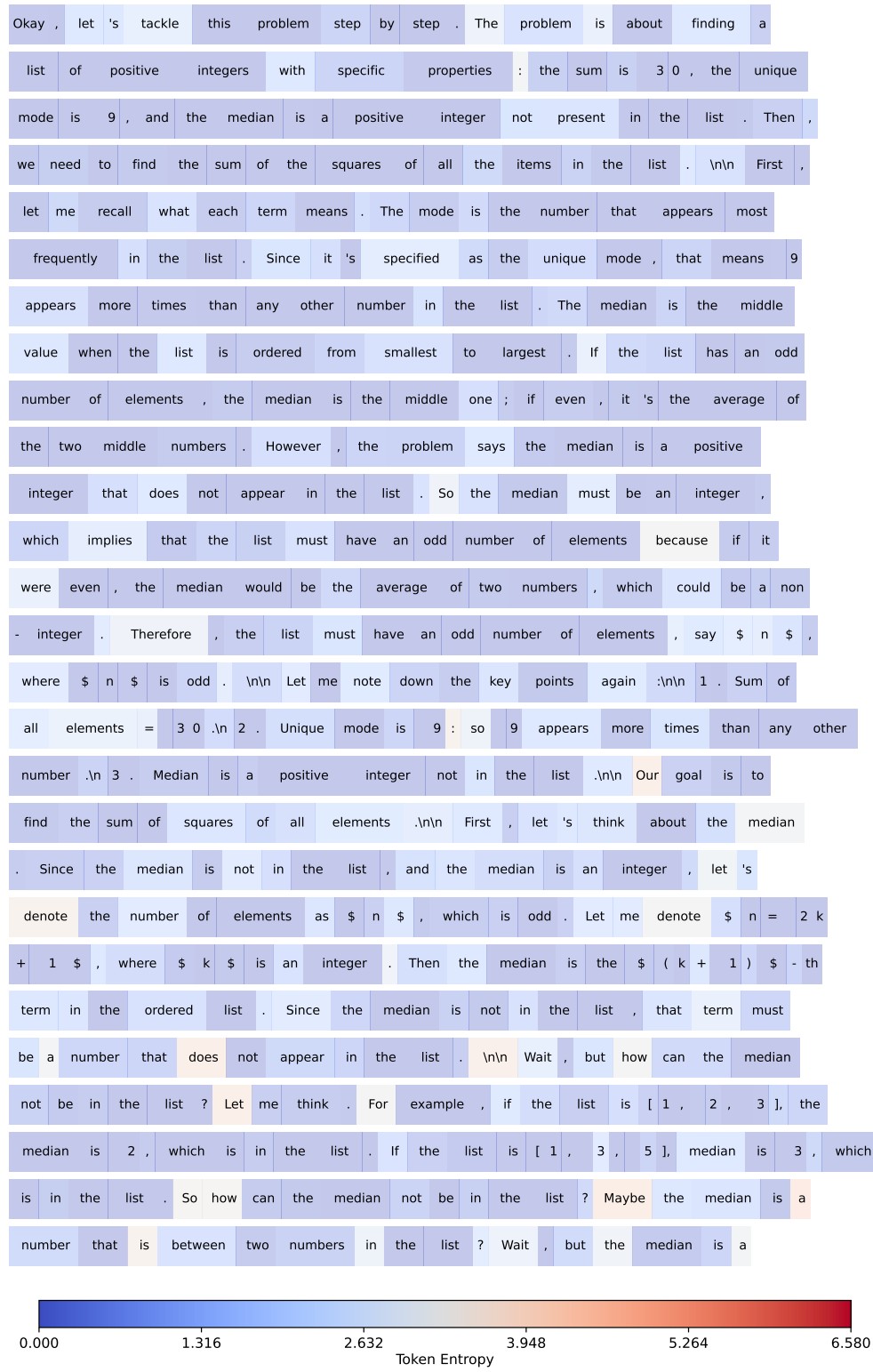

Figure 12: Visualization of token entropy (part 1).

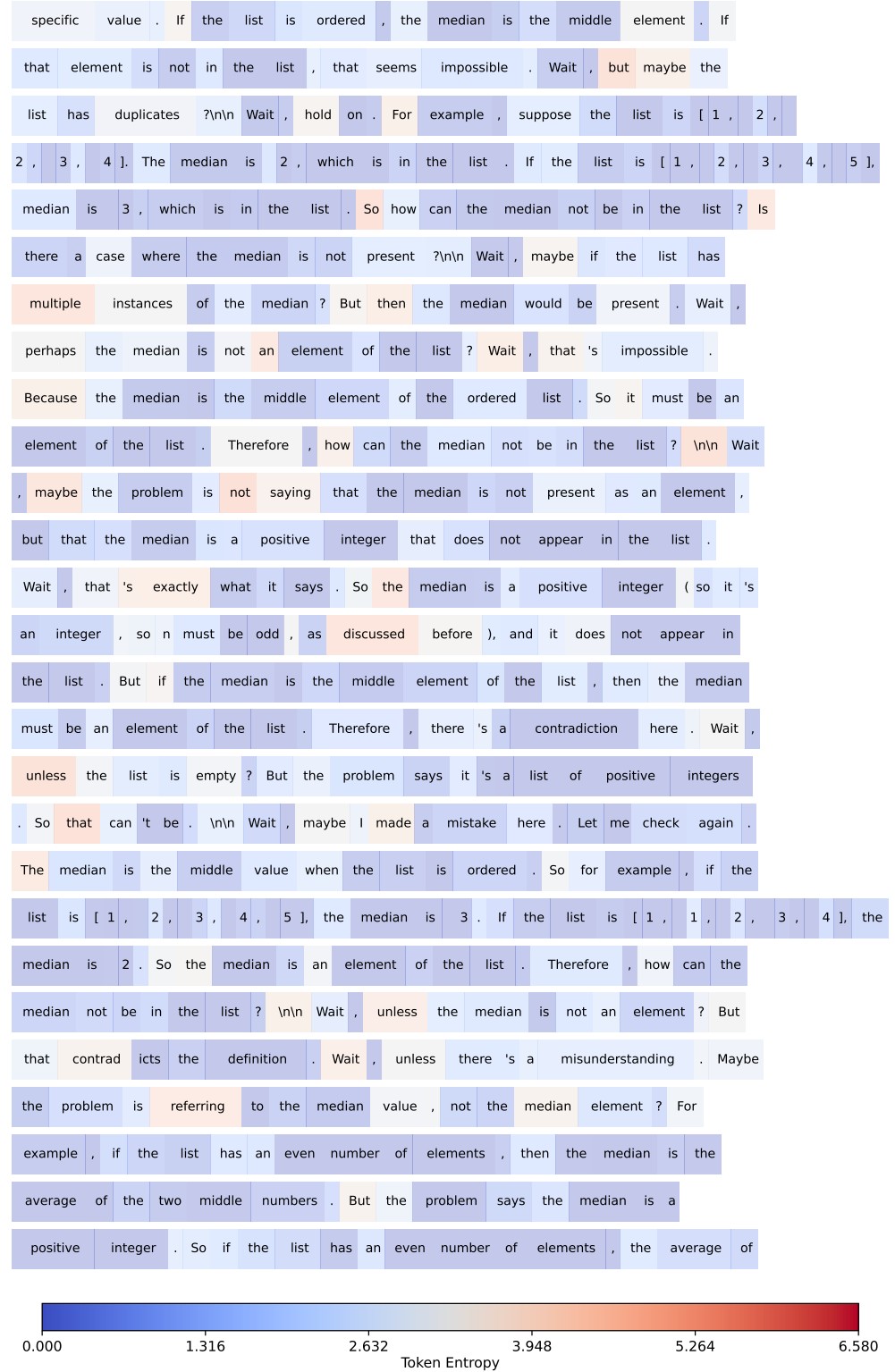

Figure 13: Visualization of token entropy (part 2).

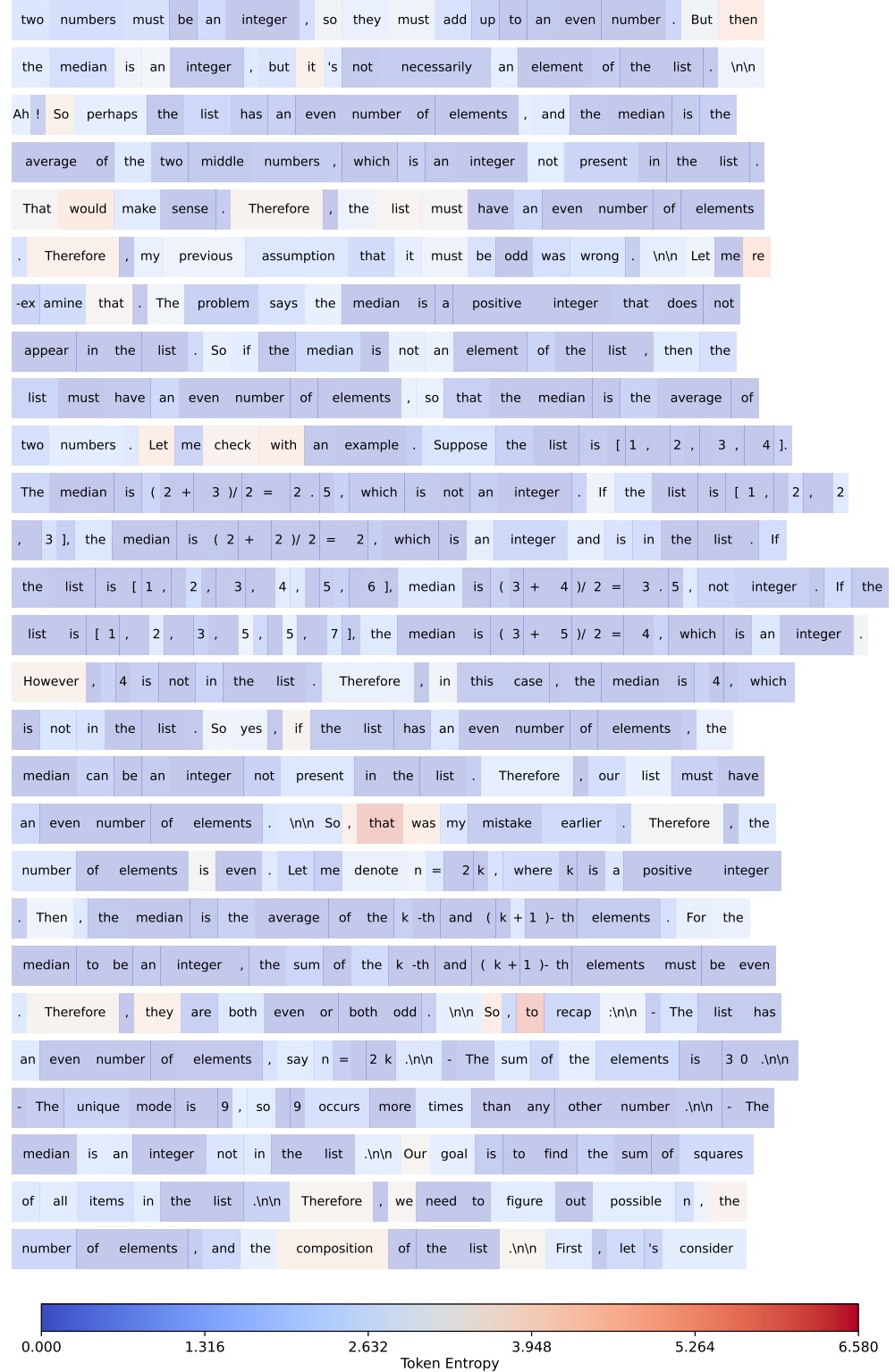

Figure 14: Visualization of token entropy (part 3).

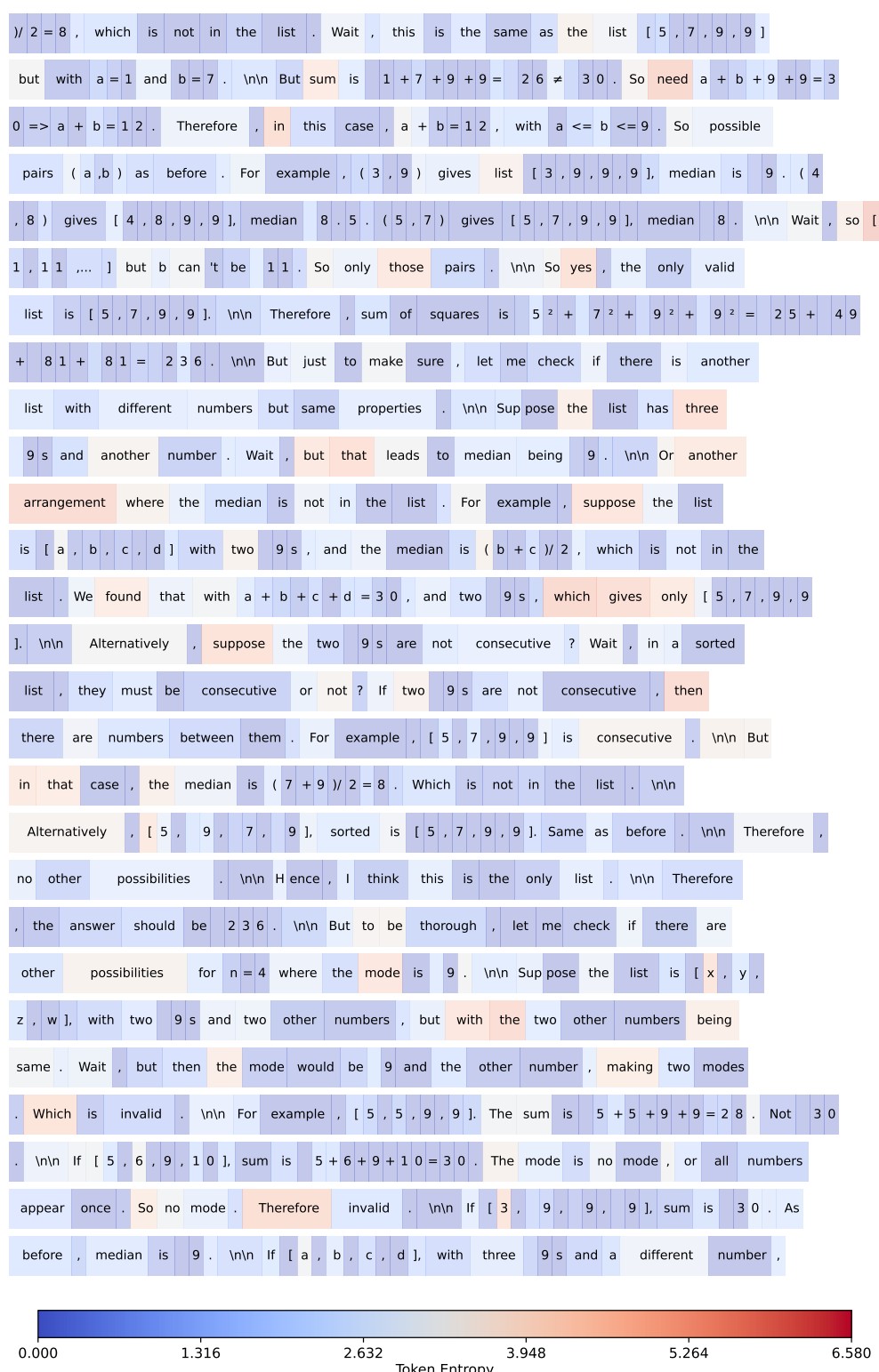

Figure 15: Visualization of token entropy (Part 4). For brevity, we omit the CoT following Figure 14 and preceding Figure 15.

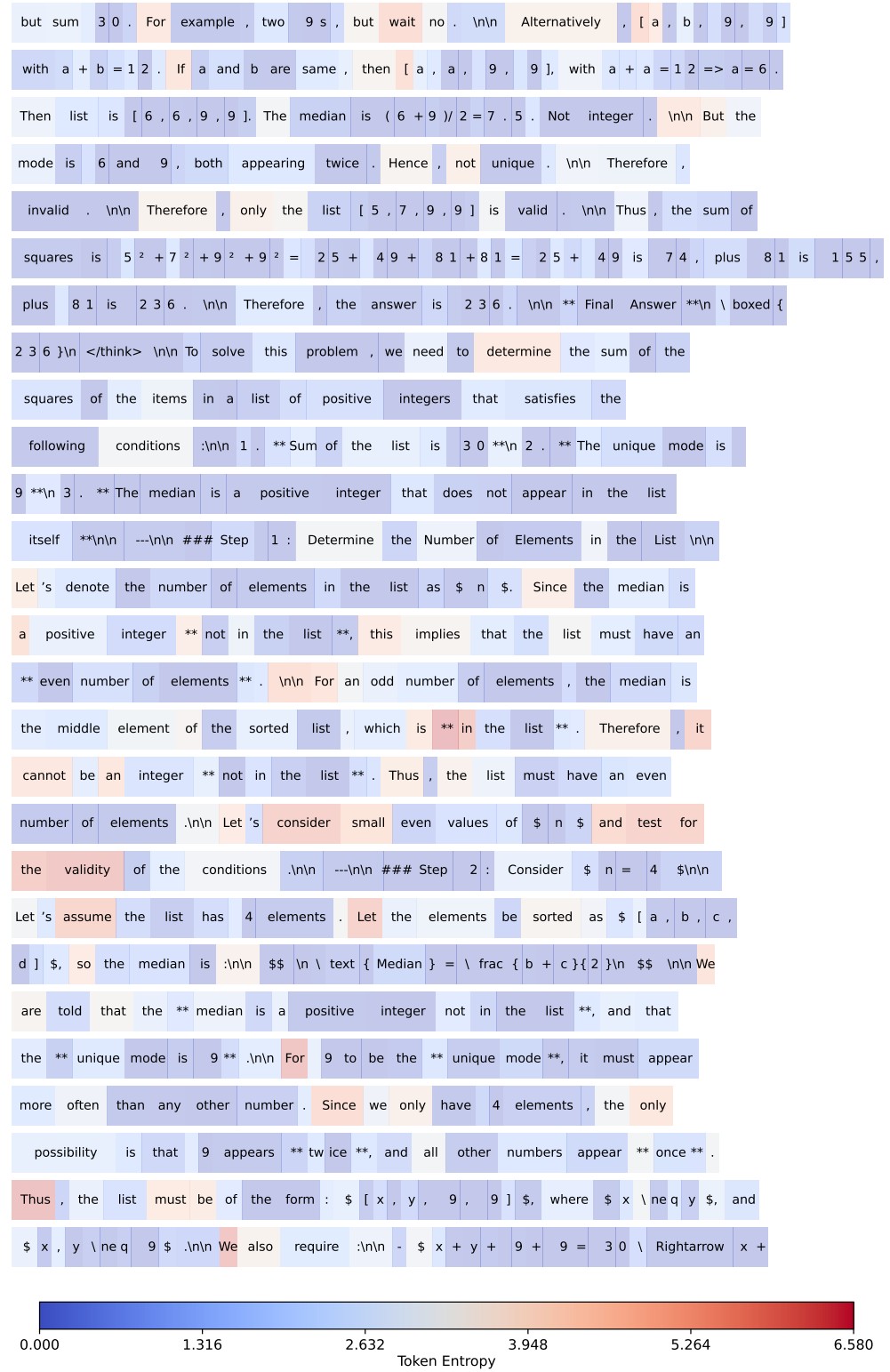

but sum 3 0 . For example , two 9 s , but wait no . \n\n Alternatively , [ a , b , 9 , 9 ] with a + b = 1 2 . If a and b are same , then [ a , a , 9 , 9 ], with a + a = 1 2 => a = 6 . Then list is [ 6 , 6 , 9 , 9 ]. The median is ( 6 + 9 )/ 2 = 7 . 5 . Not integer . \n\n But the mode is 6 and 9 , both appearing twice . Hence , not unique . \n\n Therefore , invalid . \n\n Therefore , only the list [ 5 , 7 , 9 , 9 ] is valid . \n\n Thus , the sum of squares is 5 ² + 7 ² + 9 ² + 9 ² = 2 5 + 4 9 + 8 1 + 81 = 2 5 + 4 9 is 7 4 , plus 8 1 is 1 5 5 , plus 8 1 is 2 3 6 . \n\n Therefore , the answer is 2 3 6 . \n\n ** Final Answer **\n \ boxed { 2 3 6 }\n </think> \n\n To solve this problem , we need to determine the sum of the squares of the items in a list of positive integers that satisfies the following conditions :\n\n 1 . ** Sum of the list is 3 0 **\n 2 . ** The unique mode is 9 **\n 3 . ** The median is a positive integer that does not appear in the list itself **\n\n --\n\n ### Step 1 : Determine the Number of Elements in the List \n\n Let 's denote the number of elements in the list as $ n $. Since the median is a positive integer ** not in the list **, this implies that the list must have an ** even number of elements ** . \n\n For an odd number of elements , the median is the middle element of the sorted list , which is ** in the list ** . Therefore , it cannot be an integer ** not in the list ** . Thus , the list must have an even number of elements .\n\n Let 's consider small even values of $ n $ and test for the validity of the conditions . \n\n --\n\n ### Step 2 : Consider $ n = 4 $\n\n Let 's assume the list has 4 elements . Let the elements be sorted as $ [ a , b , c , d ] $, so the median is :\n\n $$ \n \ text { Median } = \ frac { b + c }{ 2 }\n $$ \n\n We are told that the ** median is a positive integer not in the list **, and that the ** unique mode is 9 ** .\n\n For 9 to be the ** unique mode **, it must appear more often than any other number . Since we only have 4 elements , the only possibility is that 9 appears ** tw ice **, and all other numbers appear ** once ** . Thus , the list must be of the form : $ [ x , y , 9 , 9 ] $, where $ x \ ne q y $, and $ x , y \ ne q 9 $ .\n\n We also require :\n\n - $ x + y + 9 + 9 = 3 0 \ Rightarrow x +

Figure 16: Visualization of token entropy (part 5).

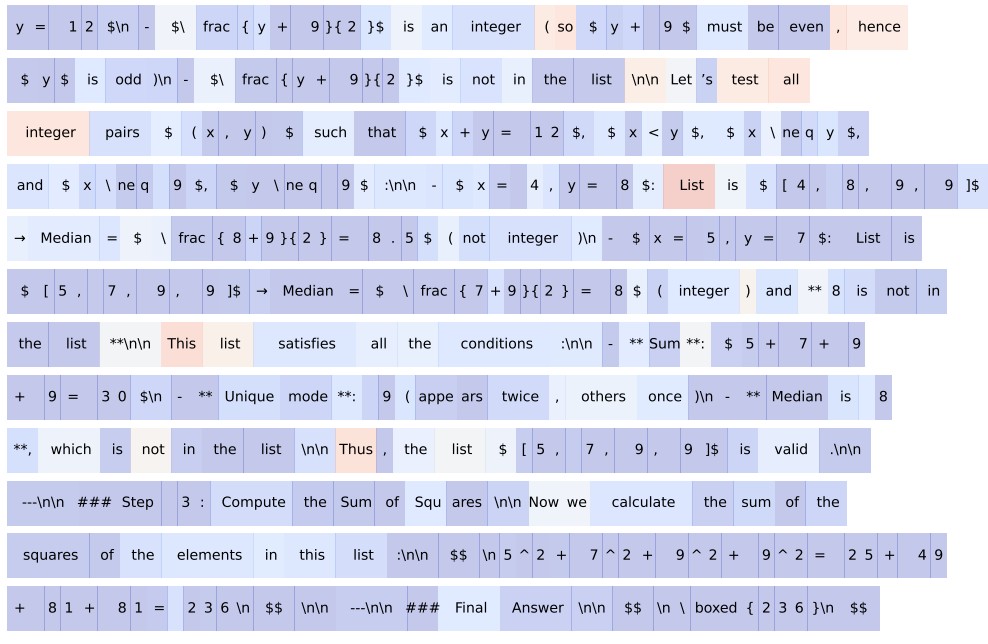

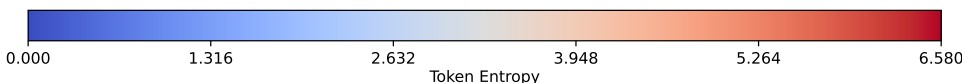

Figure 17: Visualization of token entropy (part 6).

