# OpenReview forum: "Beyond the 80/20 Rule: High-Entropy Minority Tokens Drive Effective Reinforcement Learning for LLM Reasoning"
_NeurIPS.cc/2025/Conference — NeurIPS 2025 poster_

### Official Review · Reviewer_TtMU · 2025-06-20

**Clarity:** 3
**Significance:** 3
**Originality:** 3
**Rating:** 5
**Confidence:** 5

**Summary:**

The paper analyzes the token entropy patterns in Chain-of-Thought (CoT) reasoning and observes that only a small fraction (approximately
20%) of tokens exhibit high entropy. These high-entropy tokens seem to function as "forks" to determine reasoning directions, and low-entropy tokens tend to execute reasoning steps along the established path. Based on these observations, the paper proposes to only learning on the high-entropy tokens during reinforcement learning with verifiable rewards and surpasses the baseline DAPO. The experimental results verify the importance of these high-entropy tokens.

**Questions:**

1. See weakness 1. The reviewer would expect more explanations and analysis about why the CoT length becomes longer, and how the patterns change of these high/low entropy tokens before and after RLVR on these high-entropy tokens, and why the "fork" tokens have high-entropy?
2. In the setting of ZERO training, some meaningless and garbled tokens will also have high entropy, it seems that the proposed method will encourage these garbled tokens in training, how is this issue fixed?

**Ethical Concerns:**

["NO or VERY MINOR ethics concerns only"]

**Final Justification:**

The paper addressed my concerns. According to other review comments and the corresponding responses, I tend to raise my score to 5.

**Limitations:**

See weakness 1.
The paper mainly present empirical observation and experimental results. More insights would help improve the paper.

**Quality:**

3

**Strengths And Weaknesses:**

### Strengths

1. The paper identifies an interesting phenomenon, where the tokens in the long chain-of-thought trajectories exhibit different entropy patterns, points out the difference between these tokens, and applies the analysis in RLVR.
2. The experimental results show that only learning on these high-entropy tokens can improve performance based on baseline DAPO, which is a strong baseline for RLVR.

### Weaknesses

1. The mechanism of high/low-entropy tokens lacks further analysis. For example, after RLVR on high-entropy tokens, the length of CoT becomes significantly longer. Furthermore, it is a bit counterintuitive that the tokens have high entropy on some "fork" tokens that do not exactly execute reasoning. Thus, it might be more insightful if the paper could bring deeper analysis and explanation about the high/low entropy tokens, and their corresponding changes after RLVR only on these high-entropy tokens.

---

> ### Author Rebuttal · Authors · 2025-07-31
>
> We sincerely thank you for your thorough review and valuable suggestions, which have significantly contributed to improving our manuscript.
>
> ---
>
> > Q1: The mechanism of high/low-entropy tokens lacks further analysis. For example, after RLVR on high-entropy tokens, the length of CoT becomes significantly longer. The reviewer would expect more explanations and analysis about why the CoT length becomes longer.
>
> A1: Thank you for your valuable question. We offer the following observations—both qualitative and quantitative—when comparing training using only high-entropy tokens versus training with all tokens:
>
> (1) Qualitative: In CoT reasoning, we empirically find that models trained with only the top 20% high-entropy tokens exhibit more complex logical structures compared to those trained on all tokens. This complexity includes increased occurrences of self-verification and a higher number of logical reasoning steps. (Due to space limitations in the rebuttal and the length of CoTs, we're unable to include a comparison between CoTs from full-token training and top 20% high-entropy token training. We will provide this comparison across methods in the next version of our paper.)
>
> (2) Quantitative: After convergence in RLVR training, we compute the average number of words strongly associated with self-verification and logical reasoning per CoT. The results, presented in the table below, show a significant increase in these words when training is limited to high-entropy tokens, compared to full-token training. This supports our qualitative finding: training with only high-entropy tokens leads to more complex logical reasoning, particularly with greater emphasis on self-verification.
>
> | Training Method                         | Number of "wait" per CoT | Number of "but" per CoT | Number of "however" per CoT | Number of "alternatively" per CoT |
> |----------------------------------------|---------------------------|--------------------------|------------------------------|------------------------------------|
> | Training w/ top 20% high-entropy tokens| 17.94                     | 29.13                    | 1.84                         | 3.68                               |
> | Training w/ full tokens                | 11.39                     | 11.67                    | 0.57                         | 1.49                               |
>
> **A possible explanation:** We hypothesize that this phenomenon stems from the model’s increased focus on logic-related tokens when trained on high-entropy inputs, as illustrated in Fig. 2(b). As a result, when faced with challenging problems that the model cannot initially solve, it prioritizes enhancing the logical components of the CoT to arrive at the correct answer. This process typically results in longer, more logically intricate CoTs compared to training with all tokens.
>
> ---
>
> > Q2: Furthermore, it is a bit counterintuitive that the tokens have high entropy on some "fork" tokens that do not exactly execute reasoning. Why the "fork" tokens have high-entropy?
>
> A2: Thank you for your insightful question.
>
> Regarding why "fork" tokens typically do not directly execute reasoning, this may be more intuitively understood through the metaphor illustrated in Fig. 1(a). When a person is walking along a mountain path and reaches a fork in the trail, they do not actually walk to decide which way to go. Instead, they simply change direction—turning or adjusting orientation based on a decision. Similarly, in CoT reasoning with LLMs, when the model decides on the next direction of reasoning, it doesn’t carry out the reasoning itself at that moment. Instead, it outputs logically suggestive tokens—such as “wait,” “but,” “however,” or “alternatively” (as illustrated in Fig. 2(b))—that signal a shift or branching in the reasoning path. While these tokens don’t perform the reasoning directly, they critically shape the logical flow of subsequent CoT steps.
>
> As for why the "forking" tokens have high entropy, it is because that, at positions in the CoT where multiple plausible candidate tokens exist, each plausible candidate tokens tends to have a relatively high probability. This results in a comparatively flat probability distribution. According to the token entropy definition in Equation (1):
>
> $$
> H_t = -\sum_{j=1}^{V} p_{t,j} \log p_{t,j},
> $$
>
> a flatter distribution typically yields higher entropy. This explains why "forking" tokens exhibit high entropy.
>
>
> ---
>
> > Q3: Thus, it might be more insightful if the paper could bring deeper analysis and explanation about the high/low entropy tokens, and their corresponding changes after RLVR only on these high-entropy tokens.
>
> A3: Thank you for your constructive suggestion. We conducted a deeper analysis on how token entropy evolves after RLVR, and surprisingly found that the overall entropy pattern (i.e., which tokens exhibit high entropy and which exhibit low entropy) remains largely unchanged. Specifically, we use the RL-trained LLM to generate responses, then compute the logits for these responses using both the pre-RL model (i.e., the base model) and the post-RL model (i.e., the RLVR model). We then calculate the overlap in token positions within the top 20% entropy. The results are presented in the table below.
>
> | Compared w/  | Step 0 | Step 16 | Step 112 | Step 160 | Step 480 | Step 800 | Step 864 | Step 840 | Step 1280 | Step 1360 |
> |--------------|--------|---------|----------|----------|----------|----------|----------|----------|-----------|------------|
> | Base Model   | 100%   | 98.92%  | 98.70%   | 93.04%   | 93.02%   | 93.03%   | 87.45%   | 87.22%   | 87.09%    | 86.67%     |
> | RLVR Model   | 86.67% | 86.71%  | 86.83%   | 90.64%   | 90.65%   | 90.64%   | 96.61%   | 97.07%   | 97.34%    | 100%       |
>
> Furthermore, we observe that this finding holds regardless of whether full tokens or only high-entropy tokens are used during training. The only difference is that training exclusively with high-entropy tokens amplifies the changes in entropy for those tokens, resulting in a higher overall entropy level as shown in Fig. 5.
>
> Thanks for your valuable suggestion again, and we will add the analysis above to the next version of our paper.
>
> ---
>
> > Q4: In the setting of ZERO training, some meaningless and garbled tokens will also have high entropy, it seems that the proposed method will encourage these garbled tokens in training, how is this issue fixed?
>
> A4: Thank you for your valuable feedback. We also observed that in CoT reasoning, certain seemingly meaningless or garbled tokens—such as `.`, `,`, or `\n\n`—can still exhibit high entropy. However, it is difficult to determine whether these tokens are truly meaningless or garbled. For instance, although they may appear insignificant to humans, such tokens could potentially serve as transitional elements in CoTs, helping to link preceding and subsequent thoughts.
>
> That said, we acknowledge that entropy is only a proxy metric for identifying tokens that are more valuable for training. This metric is not perfect and can sometimes include genuinely meaningless or garbled tokens in the training process. We believe exploring more effective proxy metrics for identifying important tokens is a promising direction for future research.
>
> Thank you for pointing this out, and we will incorporate this discussion into the limitations section of the next version of our paper.
>
>
> ---
>
> We sincerely appreciate your expertise and careful evaluation. **If you find our responses satisfactory, we kindly ask you to consider adjusting the score accordingly.** We welcome any further discussion and are happy to address any additional questions or concerns you may have.

---

> > ### Author Response · Authors · 2025-08-06
> > **A Kind Follow-Up Regarding Review Response**
> >
> > Dear Reviewer TtMU,
> >
> > We sincerely appreciate the time and effort you dedicated to the review process. Your insightful feedback has significantly contributed to improving our paper.
> >
> > For your convenience, we have summarized the seven additional experiments and analyses included in **[this official comment](https://openreview.net/forum?id=yfcpdY4gMP&noteId=q2IqDDGHwM)**:
> >
> > 1. Generalization to different data domains;
> > 2. Generalization to different algorithms;
> > 3. Generalization to different models;
> > 4. A deeper analysis of entropy pattern evolution during RLVR training;
> > 5. Quantitative explanation for increased response length;
> > 6. Included error bars to validate the statistical significance of the performance improvement;
> > 7. Theoretical analysis on why, under common policy parameterizations, high-entropy tokens tend to yield larger policy gradients, thereby more strongly influencing the update.
> >
> > We kindly ask whether these added experiments and theoretical insights have adequately addressed your concerns. We would be grateful if you found them valuable and chose to reflect this in your rating.
> >
> > Once again, thank you for your thoughtful and constructive review.
> >
> > Best regards,
> >
> > The authors of “High-Entropy Minority Tokens Drive Effective Reinforcement Learning for LLM Reasoning”

---

> > > ### Author Response · Authors · 2025-08-08
> > >
> > > Dear Reviewer TtMU,
> > >
> > > As there are fewer than 24 hours remaining in the rebuttal period, we would greatly appreciate it if the reviewer could confirm whether our latest response has addressed their final concerns.
> > > We are grateful for the constructive suggestions provided, which not only helped us **resolve the issues raised** but also led us to **implement 7 additional improvements, as summarized above**, significantly enhancing our manuscript.
> > >
> > > If the reviewer feels that our efforts and responses are satisfactory, we kindly ask you to consider updating your final score. Thank you for your time and dedication throughout the review process. We remain available to address any further questions or concerns.
> > >
> > > Best regards,
> > >
> > > The authors of “High-Entropy Minority Tokens Drive Effective Reinforcement Learning for LLM Reasoning”

---

### Official Review · Reviewer_q52N · 2025-07-02

**Clarity:** 3
**Significance:** 4
**Originality:** 3
**Rating:** 5
**Confidence:** 5

**Summary:**

This paper investigates the mechanisms underlying Reinforcement Learning with Verifiable Rewards (RLVR) for enhancing LLM reasoning capabilities through the lens of token-level entropy analysis. The authors demonstrate that in Chain-of-Thought reasoning, only approximately 20% of tokens exhibit high entropy and serve as critical "forking points" that determine reasoning directions, including some common tokens indicating test-time scaling like "wait". They propose a modified RLVR approach that restricts policy gradient updates to only these high-entropy tokens, achieving comparable or superior performance to full-gradient methods while using only 20% of tokens. The key finding is that low-entropy tokens contribute minimally to reasoning improvements, with the high-entropy minority tokens being the primary drivers of RLVR effectiveness.

**Questions:**

How did you determine that the high-entropy tokens are forking tokens? In the paper, this is a very strong causal claim ("We refer to high-entropy tokens as forking tokens, which function like forks in the road by determining which path to follow for continuation"), so more convincing and comprehensive evidence should be provided.

**Ethical Concerns:**

["NO or VERY MINOR ethics concerns only"]

**Final Justification:**

The authors said they will solve the writing issues in the next version, which is good. Given that I already gave them a high score and the change does not fundamentally improve paper quality, I will maintain my rating.

**Limitations:**

The experimental setups. See weakness.

**Quality:**

4

**Strengths And Weaknesses:**

- Novel analysis perspective. Although there have been recent analyses of entropy for LLM reasoning, they have mostly focused on sequence-level [1], whereas this work focuses on the token-level. As token-level entropy is adopted as the proxy to gauge entropy, the angle in this work may be more suitable.
- Strong empirical validation. To investigate the entropy issue, this paper starts with an inspection of the distribution of token entropy. To further validate the hypothesis on the existence of high/low-entropy tokens, a decoding experiment is performed, showing that high-entropy tokens are pruned to high entropy while low-entropy tokens require lower temperature. **My only complaint is that the conclusion that high-entropy tokens are "fork tokens" is mainly based on qualitative analysis through word clouds.** As this terminology has repeatedly occurred and seems to be an important concept introduced in this paper, I expect there is a more rigorous derivation and evidence.
- Useful insights in practice. This work provides valuable insights into preserving the exploration ability in RL, following previous practices [2]. Together with other recent works, it calls for attention to entropy and exploration in general during RL. Also, if the authors can more scientifically validate that high-entropy tokens are "fork tokens", this will be even more interesting/important as it helps people attribute the underlying reasons.
- The paper is well-written, and the experiments present solid improvements.
- Biggest concern: the experiments are mainly focused on Qwen models on math, which has been proven to be a "noisy" setup, as we can not disentangle whether the effectiveness comes from the base model or the algorithm itself [3,4,5].

[1] The Entropy Mechanism of Reinforcement Learning for Reasoning Language Models. Cui et al. 2025.
[2] DAPO: An Open-Source LLM Reinforcement Learning System at Scale. Yu et al. 2025.
[3] Spurious Rewards: Rethinking Training Signals in RLVR. Shao et al. 2025.
[4] The Unreasonable Effectiveness of Entropy Minimization in LLM Reasoning. Agarwal et al. 2025.
[5] Reinforcement Learning for Reasoning in Large Language Models with One Training Example. Wang et al. 2025.

---

> ### Author Rebuttal · Authors · 2025-07-31
>
> We sincerely thank you for your thorough review and valuable suggestions, which have significantly contributed to improving our manuscript.
>
> ---
>
> > Q1: My only complaint is that the conclusion that high-entropy tokens are "fork tokens" is mainly based on qualitative analysis through word clouds. As this terminology has repeatedly occurred and seems to be an important concept introduced in this paper, I expect there is a more rigorous derivation and evidence.
>
> A1: Thank you for highlighting this potential ambiguity. In our paper, we use the term "forking token" to describe high-entropy minority tokens. These tokens naturally have multiple plausible alternatives due to their high entropy, making them interchangeable at their respective positions. As such, they resemble "forks" in a decision path, where several viable candidates can lead to different CoTs.
>
> Moreover, as shown quantitatively in Fig. 3, the high-entropy characteristic of these tokens is essential—removing or diminishing this property results in a significant drop in performance. This supports the idea that the "forking" function of high-entropy minority tokens is critical: when we suppress the probabilities of all candidates except the most likely one (thereby reducing the forking effect), performance degrades markedly.
>
> Thanks again for pointing this out. We will clarify and emphasize this explanation in the next version of our paper.
>
> ---
>
> > Q2: How did you determine that the high-entropy tokens are forking tokens? In the paper, this is a very strong causal claim ("We refer to high-entropy tokens as forking tokens, which function like forks in the road by determining which path to follow for continuation"), so more convincing and comprehensive evidence should be provided.
>
> A2: Thanks for your pointing out. We make this claim based on the observations in `A1` above. We will add the explanations above to the next version of our paper to avoid potential misunderstanding.
>
> ---
>
> > Q3: Biggest concern: the experiments are mainly focused on Qwen models on math, which has been proven to be a "noisy" setup, as we can not disentangle whether the effectiveness comes from the base model or the algorithm itself.
>
> A3: Thank you for pointing out the potential issue. We'd like to clarify the following two points:
>
> (1) According to a recent study [1], which discusses or references the papers you listed [2, 3, 4], there is some degree of data contamination in the Qwen models on older benchmarks such as MATH500, AMC, and AIME'24. This may be unavoidable when pre-training on large-scale web corpora that include publicly available benchmark problems, as noted in [1].
>
> However, newer benchmarks like AIME'25 and MinervaMath are likely free from such contamination. Importantly, our method—training on the top 20% of high-entropy tokens—consistently outperforms full-token training on both the older benchmarks (MATH500, AMC, AIME'24) and the newer ones (AIME'25 and MinervaMath), demonstrating the robustness and effectiveness of our approach.
>
> (2) In addition to experiments on Qwen models, we also present results in Appendix A.4 using LLaMA-3.1-8B, which is believed to have less data contamination according to [1]. In this case as well, training on only the top 20% high-entropy tokens yields better performance than using all tokens. For your convenience, we summarize the performance comparison in the table below. We also plan to highlight this experiment more prominently in the next version of our paper.
>
> | Trained from           | Methods                          | Maximum Score on AIME'24 |
> |----------------------|----------------------------------|---------------------------|
> | Llama–3.1–8B–base     | w/ top 20% high-entropy tokens   | 10.21                     |
> | Llama–3.1–8B–base     | w/ full tokens                   | 7.92                      |
>
> [1] Wu, Mingqi, et al. "Reasoning or Memorization? Unreliable Results of Reinforcement Learning Due to Data Contamination." arXiv preprint arXiv:2507.10532 (2025).
>
> [2] Spurious Rewards: Rethinking Training Signals in RLVR. Shao et al. 2025.
>
> [3] The Unreasonable Effectiveness of Entropy Minimization in LLM Reasoning. Agarwal et al. 2025.
>
> [4] Reinforcement Learning for Reasoning in Large Language Models with One Training Example. Wang et al. 2025.
>
> ---
>
> We greatly appreciate your thoughtful evaluation and expertise. **If our responses address your concerns adequately, we would be grateful if you could consider increasing the score accordingly.** We remain open to further discussion and are more than willing to clarify any remaining questions or issues you might have.

---

> > ### Comment · Reviewer_q52N · 2025-08-02
> >
> > Thanks for the clarification, which makes the paper more self-constrained. However, it does not fundamentally improve this work, either, so I will remain my score.

---

> > > ### Author Response · Authors · 2025-08-05
> > >
> > > We would like to thank you for your time and effort throughout the entire review process. We believe that your proposed concerns significantly strengthened our paper. Thank you again for your thorough and thoughtful review.
> > >
> > > Best regards, The authors of “High-Entropy Minority Tokens Drive Effective Reinforcement Learning for LLM Reasoning”

---

### Official Review · Reviewer_g3nW · 2025-07-05

**Clarity:** 3
**Significance:** 4
**Originality:** 4
**Rating:** 5
**Confidence:** 4

**Summary:**

The work explores the mechanism behind  RLVR by zooming from sequence-level to token-level entropy in CoT traces of LLMs. An empirical analysis reveals that only about 20 % of tokens carry high entropy, and these “forking tokens” act as pivotal decision points that steer reasoning paths. Controlled decoding experiments show that selectively increasing temperature at these tokens boosts accuracy, whereas manipulating low-entropy tokens has little effect. The results suggest that RLVR’s efficacy stems almost entirely from optimizing this high-entropy minority, pointing toward more targeted and efficient training strategies for future reasoning-oriented LLMs.

**Questions:**

As discussed in weakness.

**Ethical Concerns:**

["NO or VERY MINOR ethics concerns only"]

**Final Justification:**

I have checked the reviewers-author discussion and the  response to my questions. I believe most of my concerns have been resolved. I will keep my original rating.

**Limitations:**

Yes

**Quality:**

4

**Strengths And Weaknesses:**

# Strenghts
1. This work shifts the discussion from sequence level entropy to token level patterns, revealing an actionable 20/80 split that had not been quantified before.
2. The authors conduct temperature manipulation and gradient masking studies to convincingly identify the contribution of forking tokens and reduce confounds.
3. The proposed method achieves equal-or-better accuracy with 80 % fewer gradient updates, promising for large-scale RLVR where compute is scarce.
4. The authors conduct comprehensive studies, including varying the kept token ratio multiple base model sizes, and six public math benchmarks, to support robustness claims.

# Weaknesses

1. All experiments target mathematical reasoning; it remains unclear whether the 20 % rule or performance gains hold for code, commonsense, or general reasoning tasks.
2. Results are restricted to Qwen3 variants; LLaMA or Mixtral models are not included. It would strengthen this work by demonstrating the generalization of this idea with different model families.
3. No error bars or significance tests are reported, so improvements could partly stem from stochasticity in RL runs.
4. While the idea is intuitively appealing, the work stops short of formalizing why high-entropy tokens dominate the policy gradient signal. Discussing with more insightful analysis from optimization or learning dynamics may further boost this work.

---

> ### Author Rebuttal · Authors · 2025-07-30
>
> Thank you for your careful review and constructive suggestions, which have helped us improve our manuscript.
>
> ---
>
> > Q1: All experiments target mathematical reasoning; it remains unclear whether the 20 % rule or performance gains hold for code, commonsense, or general reasoning tasks.
>
> A1: Thank you for your valuable feedback. Based on your suggestion, we compared DAPO on the Qwen3-14B base model using either all tokens or only the top 20% high-entropy tokens on a code dataset. For evaluation, we followed the methodology used in Qwen3 [1] and assessed model performance on code benchmarks including Codeforces Ratings from CodeElo [2] and LiveCodeBench (v5, 2024.10–2025.02) [3].
>
> Each method was trained for a total of 960 gradient steps. The performance comparison is presented in the table below. We observed that training with only the top 20% high-entropy tokens leads to a significant improvement on Codeforces, while achieving performance comparable to using all tokens on LiveCodeBench. These findings suggest that training with just the top 20% high-entropy tokens can deliver better or at least equivalent performance across different benchmarks compared to using full-token training.
>
> | Trained from        | Methods                          | Maximum Score on Codeforces | Maximum Score on LiveCodeBench v5 |
> |-------------------|----------------------------------|------------------------------|------------------------------------|
> | Qwen3–14B–Base     | w/ top 20% high-entropy tokens   | 1387                         | 30.36                              |
> | Qwen3–14B–Base     | w/ full tokens                   | 1259                         | 30.03                              |
>
> [1] Yang, An, et al. "Qwen3 technical report." arXiv preprint arXiv:2505.09388 (2025).
>
> [2] Quan, Shanghaoran, et al. "Codeelo: Benchmarking competition-level code generation of llms with human-comparable elo ratings." arXiv preprint arXiv:2501.01257 (2025).
>
> [3] Jain, Naman, et al. "Livecodebench: Holistic and contamination free evaluation of large language models for code." arXiv preprint arXiv:2403.07974 (2024).
>
> ---
>
> > Q2: Results are restricted to Qwen3 variants; LLaMA or Mixtral models are not included. It would strengthen this work by demonstrating the generalization of this idea with different model families.
>
> A2: Thank you for your helpful suggestion. We have already included experiments on LLaMA-3.1-8B in Appendix A.4, where training with only the top 20% high-entropy tokens also demonstrates an advantage over training with all tokens. For your convenience, we summarize the performance comparison in the table below. We will also make this experiment more prominent in the next version of our paper.
>
> | Trained from           | Methods                          | Maximum Score on AIME'24 |
> |----------------------|----------------------------------|---------------------------|
> | Llama–3.1–8B–base     | w/ top 20% high-entropy tokens   | 10.21                     |
> | Llama–3.1–8B–base     | w/ full tokens                   | 7.92                      |
>
> Furthermore, to verify that training with the top 20% high-entropy tokens also provides an advantage on other types of LLMs, we conducted additional comparison experiments using the Qwen2.5-7B-base and Qwen2.5-7B-instruct models. The results, shown in the table below, indicate that training with only the top 20% high-entropy tokens consistently outperforms training with all tokens.
>
> | Trained from             | Methods                          | Maximum Score on AIME'24 | Maximum Score on AIME'25 |
> |---|---|---|-------|
> | Qwen2.5–7B–Base        | w/ top 20% high-entropy tokens   | 20.21                     | 16.25                     |
> | Qwen2.5–7B–Base        | w/ full tokens                   | 19.79                     | 15.42                     |
> | Qwen2.5–7B–Instruct    | w/ top 20% high-entropy tokens   | 18.33                     | 17.50                     |
> | Qwen2.5–7B–Instruct    | w/ full tokens                   | 17.29                     | 16.25                     |
>
> ---
>
> > Q3: No error bars or significance tests are reported, so improvements could partly stem from stochasticity in RL runs.
>
> A3: Thank you for your feedback. As shown in Appendix B, each experiment requires substantial computational resources, making it difficult to run multiple trials with different random seeds. For instance, a full RLVR run on the Qwen3-32B model demands 64 A100 80GB GPUs for approximately 220 hours. Given the computational constraint, it is standard practice to perform only a single run per setting, as noted in references [1–5] mentioned by Reviewer q52N.
>
> However, we can quantify variability in an alternative way: by computing error bars based on the test scores from 11 mini-batches surrounding the peak score (five before and five after). We calculate the average and standard deviation of these scores, and the results are as follows. They indicate that, for the 32B and 14B models, training with only the top 20% of high-entropy tokens yields a statistically significant improvement over using all tokens. For the 8B model, however, the improvement is not statistically significant. This observation aligns with the claim made in our paper.
>
> | Model                                      | avg score ± std     |
> |---|----|
> | 32B w/ top 20% high-entropy tokens         | 59.00 ± 1.07         |
> | 32B w/ full tokens                         | 53.42 ± 1.17         |
> | 14B w/ top 20% high-entropy tokens         | 49.33 ± 1.79         |
> | 14B w/ full tokens                         | 42.17 ± 1.74         |
> | 8B w/ top 20% high-entropy tokens          | 32.31 ± 1.58         |
> | 8B w/ full tokens                          | 31.28 ± 1.23         |
>
> [1] DAPO: An Open-Source LLM Reinforcement Learning System at Scale. Yu et al. 2025.
>
> [2] The Entropy Mechanism of Reinforcement Learning for Reasoning Language Models. Cui et al. 2025.
>
> [3] Spurious Rewards: Rethinking Training Signals in RLVR. Shao et al. 2025.
>
> [4] The Unreasonable Effectiveness of Entropy Minimization in LLM Reasoning. Agarwal et al. 2025.
>
> [5] Reinforcement Learning for Reasoning in Large Language Models with One Training Example. Wang et al. 2025.
>
> ---
>
> > Q4: While the idea is intuitively appealing, the work stops short of formalizing why high-entropy tokens dominate the policy gradient signal. Discussing with more insightful analysis from optimization or learning dynamics may further boost this work.
>
> A4: Thank you for the insightful comment. We clarify here why, under common policy parameterizations, high-entropy tokens tend to yield larger policy gradients, thereby more strongly influencing the update.
>
> Recall the (unclipped) policy gradient term for a single sample:
>
> $$
> g_t = A_t \cdot r_\theta(t) \cdot \nabla_\theta \log \pi_\theta(a_t \mid s_t),
> \quad \text{where } r_\theta(t) = \frac{\pi_\theta(a_t \mid s_t)}{\pi_{\text{old}}(a_t \mid s_t)}.
> $$
>
> While all three terms affect the gradient magnitude, in practice:
>
> (1) The advantage estimate $A_t$ is treated as fixed during optimization (precomputed from rollout), and is often normalized (i.e., normalized within each group in DAPO or GRPO) per update.
> (2) The importance ratio $r_\theta(t)$ affects the gradient only multiplicatively, and in DAPO is further clipped to $[1 - \epsilon_{low}, 1 + \epsilon_{high}]$, limiting its variability.
> (3) The score function $\nabla_\theta \log \pi_\theta(a_t \mid s_t)$ is the only term directly affected by the policy entropy and dominates the direction and magnitude of the update.
>
> Specifically, under a softmax policy for discrete actions, assume the last network layer outputs a logit vector $z\in\mathbb R^{|A|}$.
> The soft-max policy is
>
> $$
> p_i =\pi_{\theta}(a=i | s) = \frac{\exp(z_i)}{\sum_{j}\exp(z_j)}, \quad i=1,2,\cdots, |A|.
> $$
>
> For a sampled action $a$ the log-probability is
>
> $$
> \log \pi_\theta(a\mid s)= z_{a}-\log\bigl(\textstyle\sum_j\exp(z_j)\bigr).
> $$
>
> Gradient w.r.t. the logits.  Denote $g_k=\partial\log\pi/\partial z_k$:
>
> $$
> g_k = \delta_{ka}-p_k,
> \qquad k=1,\dots,|A|.
> $$
>
> Hence for the component $k=a$
>
> $$
> g_{a}=1-p_{a},
> $$
>
> while for every $k\neq a$
>
> $$
> g_k=-p_k.
> $$
>
> Magnitude for one sample.  The (squared) Euclidean norm of this sample gradient is
>
> $$
> \lVert \nabla_{z}\log\pi_\theta(a\mid s)\rVert_2^2
> =\sum_{k}(g_k)^2
> =(1-p_a)^2+\sum_{k\neq a}p_k^{\,2}.
> $$
>
> What happens at different entropy levels?
>
> - Near-deterministic policy, e.g. $p_a\approx1$: $\quad  1-p_a\approx0$ and all other $p_k\approx0$  $\Rightarrow$  gradient nearly vanishes.
>
> - High-entropy (near-uniform) policy, $p_i\approx 1/|A|$: $\quad$  All terms stay away from 0  $\Rightarrow$  gradient norm reaches its maximum.
>
> Hence a soft-max policy automatically produces larger raw policy gradients when its output distribution is high-entropy and smaller gradients when it becomes low-entropy.
>
> Although our primary interest lies in $\nabla_\theta \log \pi_\theta(a \mid s)$, our analysis focuses on $\nabla_z \log \pi_\theta(a \mid s)$, justified by the chain rule:
>
> $$
> \nabla_\theta \log \pi_\theta(a \mid s) = \nabla_z \log \pi_\theta(a \mid s) \cdot \nabla_\theta z(s; \theta).
> $$
>
> This expression highlights that the structure and magnitude of the score function in parameter space are shaped by the gradient with respect to logits. While the Jacobian $\nabla_\theta z$ is architecture-dependent, $\nabla_z \log \pi$ reflects how the output distribution’s shape — especially its entropy — affects the gradient signal. Under mild assumptions on the Jacobian norm, $\nabla_\theta \log \pi$ and $\nabla_z \log \pi$ remain aligned through backpropagation, making the latter a valid and informative proxy for studying entropy's influence on policy gradient magnitude.
>
> ---
>
> Thank you again for your expertise and thoughtful review. **If you find our response helpful, please consider raising the score.** We're happy to continue the discussion and address any further questions.

---

### Official Review · Reviewer_2VsV · 2025-07-06

**Clarity:** 4
**Significance:** 3
**Originality:** 3
**Rating:** 4
**Confidence:** 3

**Summary:**

This paper investigates the underlying mechanisms of Reinforcement Learning with Verifiable Rewards (RLVR) through the lens of token entropy patterns in reasoning models. The authors analyze token entropy distributions in LLM reasoning processes and identify that approximately 20% of tokens exhibit high entropy and function as "forking tokens" that determine reasoning directions. They propose restricting policy gradient updates to only these high-entropy tokens during RLVR training. Experiments on Qwen3-8B and Qwen3-32B models across mathematical reasoning benchmarks show that this selective approach can match or exceed full-gradient RLVR performance while using only 20% of tokens.

**Questions:**

- Simple experiments on GRPO or PPO may help clarify whether the proposed method can generalized to other RL algorithms. (If not, what is the reason?)
- The response length is also increasing with the improved accuracy. How does this phenomenon related to the training of High-Entropy Minority Tokens?

**Ethical Concerns:**

["NO or VERY MINOR ethics concerns only"]

**Limitations:**

yes

**Quality:**

3

**Strengths And Weaknesses:**

Strengths
- Novel analytical perspective: The token-level entropy analysis of RLVR mechanisms provides a fresh and interpretable lens for understanding how reinforcement learning improves reasoning capabilities. The identification of "forking tokens" as critical decision points is intuitive and well-motivated.
- Solid empirical validation: The paper provides multiple lines of evidence supporting the importance of high-entropy tokens.
- Practical significance and Simplicity: The finding that RLVR effectiveness can be improved while reducing computational requirements has practical implications for both performance and training efficiency.
- Comprehensive experimental design: The evaluation spans multiple mathematical reasoning benchmarks and includes appropriate ablations varying token proportions.

Weaknesses
- Lack of experiments on GRPO and PPO: The experiments are mainly conducted on DAPO. It will be helpful to also include experiments on GRPO or PPO to see if the proposed approach generalize to other RL algorithms.

---

> ### Author Rebuttal · Authors · 2025-07-29
>
> Thank you for your thoughtful review and valuable suggestions, which have contributed to improving our manuscript.
>
> ---
>
> > Q1: Simple experiments on GRPO or PPO may help clarify whether the proposed method can generalized to other RL algorithms. (If not, what is the reason?)
>
> A1: Thank you for the insightful suggestion. We selected DAPO over GRPO as the baseline because DAPO is an enhanced version of GRPO, incorporating four techniques (i.e., clip-higher, dynamic sampling, token-level policy gradient loss, and overlong reward shaping) that strengthen training stability in RL and substantially improve model performance. As a result, DAPO serves as a more robust and informative baseline for comparison.
>
> As for PPO, it relies on a separate critic model, which increases VRAM usage and adds complexity to the training process. For these reasons, many recent state-of-the-art reasoning models [1–4] have adopted critic-free RL methods such as GRPO and DAPO.
>
> Considering both of these factors, we chose DAPO over GRPO or PPO as our baseline.
>
> However, we agree that including an additional baseline can provide more insight into whether the proposed method generalizes to other RL algorithms. To this end, we removed the clip-higher, dynamic sampling, and token-level policy gradient loss techniques from DAPO to revert to the GRPO algorithm. We then compared training using only the top 20% high-entropy tokens against training with all tokens, using the Qwen3-14B base model. The results, shown in the table below, indicate that even with the GRPO algorithm, training on the top 20% high-entropy tokens still significantly outperforms training on the full set of tokens.
>
> | # of Gradient Steps     | AIME'24 (w/ top 20% high-entropy tokens) | AIME'24 (w/ full tokens) | AIME'25 (w/ top 20% high-entropy tokens) | AIME'25 (w/ full tokens) |
> |----------|------------------------------------------|---------------------------|------------------------------------------|---------------------------|
> | step 160 | 23.96                                    | 23.54                     | 25.83                                    | 31.25                     |
> | step 320 | 26.25                                    | 14.58                     | 29.58                                    | 15.83                     |
> | step 480 | 26.46                                    | 19.38                     | 30.83                                    | 22.50                     |
> | step 640 | 28.33                                    | 18.54                     | 31.67                                    | 12.08                     |
> | step 720 | 30.21                                    | 16.67                     | 32.92                                    | 18.75                     |
> | step 800 | 27.50                                    | 19.79                     | 32.92                                    | 18.75                     |
> | step 960 | 27.50                                    | 17.50                     | 33.33                                    | 20.00                     |
>
> [1] Guo, Daya, et al. "Deepseek-r1: Incentivizing reasoning capability in llms via reinforcement learning." arXiv preprint arXiv:2501.12948 (2025).
>
> [2] Yang, An, et al. "Qwen3 technical report." arXiv preprint arXiv:2505.09388 (2025).
>
> [3] Chen, Aili, et al. "MiniMax-M1: Scaling Test-Time Compute Efficiently with Lightning Attention." arXiv preprint arXiv:2506.13585 (2025).
>
> [4] Rastogi, Abhinav, et al. "Magistral." arXiv preprint arXiv:2506.10910 (2025).
>
> -------------
>
> > Q2: The response length is also increasing with the improved accuracy. How does this phenomenon related to the training of High-Entropy Minority Tokens?
>
> A2: Thank you for your insightful observation. We have found through experiments that focusing training only on high-entropy minority tokens—specifically the forking tokens shown in Fig. 2(b), which often act as logical connectors within and between sentences—can make the chain-of-thought (CoT) logic more complex, especially with more self-verifications and logical turns. As a result, response length tends to grow alongside this more complex logic.
>
> Due to the character limits of the rebuttal response and long length of CoT, we cannot provide comparison of CoTs of training with all tokens and with only top 20% high-entropy tokens. We will provide a comparision of CoTs of different methods in the next version of our paper.
>
> To further support the qualitative observation above, we include quantitative results. After convergence in RLVR training, we calculate the average number of words strongly associated with self-verification and logical reasoning per CoT. As shown in the table below, these words appear significantly more frequently when training is restricted to high-entropy tokens compared to using all tokens. This reinforces our qualitative finding: training with only high-entropy tokens fosters more complex logical reasoning, with a particular emphasis on self-verification.
>
> | Training Method                         | Number of "wait" per CoT | Number of "but" per CoT | Number of "however" per CoT | Number of "alternatively" per CoT |
> |----------------------------------------|---------------------------|--------------------------|------------------------------|------------------------------------|
> | Training w/ top 20% high-entropy tokens| 17.94                     | 29.13                    | 1.84                         | 3.68                               |
> | Training w/ full tokens                | 11.39                     | 11.67                    | 0.57                         | 1.49                               |
>
> ---
>
> We sincerely appreciate your expertise and careful evaluation. **If you find our responses satisfactory, we kindly ask you to consider adjusting the score accordingly.** We welcome any further discussion and are happy to address any additional questions or concerns you may have.

---

> > ### Comment · Reviewer_2VsV · 2025-08-05
> >
> > I appreciate the response and additional experiments from the authors. I will keep my score for incline for acceptance.

---

> > > ### Author Response · Authors · 2025-08-06
> > > **A Kind Follow-Up Regarding Review Response**
> > >
> > > Dear Reviewer 2VsV,
> > >
> > > Thank you for your recognition of our response. We truly appreciate the time and effort you have devoted to reviewing our paper. Your thoughtful feedback has been invaluable in helping us improve our work.
> > >
> > > To address your comments thoroughly and further improve our work, we have invested significant effort to strengthen the paper both empirically and theoretically. A summary of the seven additional experiments and analyses included in **[this response](https://openreview.net/forum?id=yfcpdY4gMP&noteId=q2IqDDGHwM)** is as follows:
> > >
> > > 1. Generalization to different data domains;
> > > 2. Generalization to different algorithms;
> > > 3. Generalization to different models;
> > > 4. A deeper analysis of entropy pattern evolution during RLVR training;
> > > 5. Quantitative explanation for increased response length;
> > > 6. Inclusion of error bars to assess the statistical significance of the observed performance gains;
> > > 7. Theoretical analysis explaining why, under common policy parameterizations, high-entropy tokens tend to yield larger policy gradients and thus play a more influential role during updates.
> > >
> > > We sincerely hope that these additional efforts have addressed your concerns. **If you find the revisions and updates meaningful, we would be truly grateful if you would consider reflecting that in your final evaluation.**
> > >
> > > Once again, thank you very much for your detailed and constructive review.
> > >
> > > The authors of “High-Entropy Minority Tokens Drive Effective Reinforcement Learning for LLM Reasoning”

---

> > > > ### Author Response · Authors · 2025-08-08
> > > >
> > > > Dear Reviewer 2VsV,
> > > >
> > > > With less than 24 hours remaining in the rebuttal phase, we would sincerely appreciate it if the reviewer could let us know whether our most recent response has adequately addressed their remaining concerns.
> > > >
> > > > We are thankful for the reviewer’s constructive feedback, which not only allowed us to **effectively resolve the identified issues** but also **motivated us to incorporate 7 additional enhancements, as outlined above**, to further strengthen our manuscript.
> > > >
> > > > If the reviewer finds our revisions and responses satisfactory, we kindly request that you consider updating your final score. We truly appreciate your time and effort throughout the review process and are happy to respond to any further questions or concerns you may have.
> > > >
> > > > Best regards,
> > > >
> > > > The authors of “High-Entropy Minority Tokens Drive Effective Reinforcement Learning for LLM Reasoning”

---

### Comment · Area_Chair_zyD6 · 2025-08-04
**Engage in Author-Reviewer Discussions**

Dear reviewers,

If you haven't done so already, please click the 'Mandatory Acknowledgement' button and actively participate in the rebuttal discussion with the authors after carefully reading all other reviews and the author responses.

Thanks,
AC

---

### Author Response · Authors · 2025-08-05
**[1/2] A Summary of Additional Experimental Results and Analyses during the Rebuttal Phase**

Dear AC and reviewers,

Thank you for taking the time to review our paper. For your convenience, we provide a summary of the additional experimental results and analyses conducted during the rebuttal phase. **We hope these additions further strengthen our work and encourage you to consider updating your evaluation accordingly.**


---

1. **Generalization to different data domains**: In addition to training on **math data**, we also trained on **code data** to verify that the performance gain from using only the top 20% high-entropy tokens extends to other domains.

| Train from         | Methods                          | Maximum Score on Codeforces | Maximum Score on LiveCodeBench v5 |
|---|---|---|---|
| Qwen3–14B–Base     | w/ top 20% high-entropy tokens   | 1387                         | 30.36                              |
| Qwen3–14B–Base     | w/ full tokens                   | 1259                         | 30.03                              |

---

2. **Generalization across algorithms**: Beyond the **DAPO** algorithm, we incorporated the **GRPO** algorithm as an additional baseline to further demonstrate that the advantage of training with only the top 20% high-entropy tokens generalizes to other reinforcement learning methods.

Comparison using the GRPO algorithm with the Qwen3-14B base model:

| Step     | AIME'24 w/ Top 20% High-Entropy Tokens | AIME'24 w/ Full Tokens | AIME'25 w/ Top 20% High-Entropy Tokens | AIME'25 w/ Full Tokens |
|---|---|---|---|---|
| step 160 | 23.96                                  | 23.54                  | 25.83                                  | 31.25                  |
| step 320 | 26.25                                  | 14.58                  | 29.58                                  | 15.83                  |
| step 480 | 26.46                                  | 19.38                  | 30.83                                  | 22.50                  |
| step 640 | 28.33                                  | 18.54                  | 31.67                                  | 12.08                  |
| step 720 | 30.21                                  | 16.67                  | 32.92                                  | 18.75                  |
| step 800 | 27.50                                  | 19.79                  | 32.92                                  | 18.75                  |
| step 960 | 27.50                                  | 17.50                  | 33.33                                  | 20.00                  |

---

3. **Generalization across different models**: In addition to testing with **Qwen3 variants**, we included **Qwen2.5 (both base and instruct versions) and Llama 3.1 variants** to further validate that the benefits of training with only the top 20% high-entropy tokens apply to different models.

| Train from          | Methods                         | Maximum Score on AIME'24 | Maximum Score on AIME'25 |
|---|---|---|---|
| Qwen2.5–7B–Base     | w/ top 20% high-entropy tokens   | 20.21                     | 16.25                     |
| Qwen2.5–7B–Base     | w/ full tokens                   | 19.79                     | 15.42                     |
| Qwen2.5–7B–Instruct | w/ top 20% high-entropy tokens   | 18.33                     | 17.50                     |
| Qwen2.5–7B–Instruct | w/ full tokens                   | 17.29                     | 16.25                     |

Additional results (also available in Appendix A.4 of our manuscript):

| Train from          | Methods                         | Maximum Score on AIME'24 |
|---|---|---|
| Llama–3.1–8B–base   | w/ top 20% high-entropy tokens   | 10.21                     |
| Llama–3.1–8B–base   | w/ full tokens                   | 7.92                      |

---

4. **A deeper analysis of entropy pattern evolution during RLVR training**: We surprisingly observe that **the overall entropy pattern—i.e., which tokens have high entropy and which have low entropy—remains largely consistent throughout RLVR training**. Specifically, we generate responses using the RL-trained LLM and compute logits for these responses using both the pre-RL model (the base model) and the post-RL model (the RLVR model). We then measure the overlap in token positions that fall within the top 20% in terms of entropy. The corresponding results are shown in the table below.

| Compared w/   | Step 0 | Step 16 | Step 112 | Step 160 | Step 480 | Step 800 | Step 864 | Step 840 | Step 1280 | Step 1360 |
|----|----|---|---|---|---|---|---|---|---|---|
| Base Model    | 100%   | 98.92%  | 98.70%   | 93.04%   | 93.02%   | 93.03%   | 87.45%   | 87.22%   | 87.09%    | 86.67%    |
| RLVR Model    | 86.67% | 86.71%  | 86.83%   | 90.64%   | 90.65%   | 90.64%   | 96.61%   | 97.07%   | 97.34%    | 100%      |

---

> ### Author Response · Authors · 2025-08-05
> **[2/2] A Summary of Additional Experimental Results and Analyses during the Rebuttal Phase**
>
> 5. **Quantitative explanation for increased response length:** After RLVR convergence, we compute the average number of words tied to self-verification and logical reasoning per CoT. As shown below, these counts rise significantly when training is limited to high-entropy tokens, supporting our observation that such training promotes more complex reasoning and longer responses.
>
> | Training Method                         | Number of "wait" per CoT | Number of "but" per CoT | Number of "however" per CoT | Number of "alternatively" per CoT |
> |--|--|--|--|--|
> | Training w/ top 20% high-entropy tokens| 17.94                     | 29.13                    | 1.84                         | 3.68                              |
> | Training w/ full tokens                | 11.39                     | 11.67                    | 0.57                         | 1.49                              |
>
> ---
>
> 6. **Included error bars to validate the statistical significance of the performance improvement**: By computing error bars based on the test scores from 11 mini-batches surrounding the peak score (five before and five after). We calculate the average and standard deviation of these scores, and the results are as follows.
>
> | Model                                 | avg score ± std     |
> |--|--|
> | 32B w/ top 20% high-entropy tokens   | 59.00 ± 1.07         |
> | 32B w/ full tokens                   | 53.42 ± 1.17         |
> | 14B w/ top 20% high-entropy tokens   | 49.33 ± 1.79         |
> | 14B w/ full tokens                   | 42.17 ± 1.74         |
> | 8B w/ top 20% high-entropy tokens    | 32.31 ± 1.58         |
> | 8B w/ full tokens                    | 31.28 ± 1.23         |
>
> ---
>
> 7. **Theoretical analysis on why, under common policy parameterizations, high-entropy tokens tend to yield larger policy gradients, thereby more strongly influencing the update.**
>
> Recall the (unclipped) policy gradient term for a single sample:
>
> $$g_t = A_t \cdot r_\theta(t) \cdot \nabla_\theta \log \pi_\theta(a_t \mid s_t), \quad \text{where } r_\theta(t) = \frac{\pi_\theta(a_t \mid s_t)}{\pi_{\text{old}}(a_t \mid s_t)}.$$
>
> While all three terms affect the gradient magnitude, in practice:
>
> 1. The advantage estimate $ A_t $ is treated as fixed during optimization (precomputed from rollout), and is often normalized (i.e., normalized within each group in DAPO or GRPO) per update.
> 2. The importance ratio $ r_\theta(t) $ affects the gradient only multiplicatively, and in DAPO is further clipped to $ [1 - \epsilon_{\text{low}}, 1 + \epsilon_{\text{high}}] $, limiting its variability.
> 3. The score function $ \nabla_\theta \log \pi_\theta(a_t \mid s_t) $ is the only term directly affected by the policy entropy and dominates the direction and magnitude of the update.
>
> Specifically, under a softmax policy for discrete actions, assume the last network layer outputs a logit vector $ z \in \mathbb{R}^{|A|} $. The soft-max policy is:
>
> $$p_i = \pi_\theta(a = i \mid s) = \frac{\exp(z_i)}{\sum_j \exp(z_j)}, \quad i = 1, 2, \ldots, |A|.$$
>
> For a sampled action $ a $, the log-probability is:
>
> $$\log \pi_\theta(a \mid s) = z_a - \log \left( \sum_j \exp(z_j) \right).$$
>
> Gradient w.r.t. the logits. Denote $ g_k \equiv \partial \log \pi / \partial z_k $:
>
> $$g_k = \delta_{ka} - p_k, \quad k = 1, \ldots, |A|.$$
>
> Hence for the component $ k = a $:
>
> $$g_a = 1 - p_a,$$
>
> while for every $ k \ne a $:
>
> $$g_k = -p_k.$$
>
> Magnitude for one sample. The (squared) Euclidean norm of this sample gradient is:
>
> $$ \| \nabla_z \log \pi_\theta(a \mid s) \|_2^2 = \sum_k (g_k)^2 = (1 - p_a)^2 + \sum \mathbb{1}[k \ne a] p_k^2. $$
>
> What happens at different entropy levels?
>
> - Near-deterministic policy, e.g. $ p_a \approx 1 $: $ 1 - p_a \approx 0 $ and all other $ p_k \approx 0 \Rightarrow $ gradient nearly vanishes.
> - High-entropy (near-uniform) policy, $ p_i \approx 1/|A| $: All terms stay away from 0 $ \Rightarrow $ gradient norm reaches its maximum.
>
> Hence a soft-max policy automatically produces larger raw policy gradients when its output distribution is high-entropy and smaller gradients when it becomes low-entropy.
>
> Although our primary interest lies in $ \nabla_\theta \log \pi_\theta(a \mid s) $, our analysis focuses on $ \nabla_z \log \pi_\theta(a \mid s) $, justified by the chain rule:
>
> $$ \nabla_\theta \log \pi_\theta(a \mid s) = \nabla_z \log \pi_\theta(a \mid s) \cdot \nabla_\theta z(s; \theta). $$
>
> This expression highlights that the structure and magnitude of the score function in parameter space are shaped by the gradient with respect to logits. While the Jacobian $ \nabla_\theta z $ is architecture-dependent, $ \nabla_z \log \pi $ reflects how the output distribution’s shape — especially its entropy — affects the gradient signal. Under mild assumptions on the Jacobian norm, $ \nabla_\theta \log \pi $ and $ \nabla_z \log \pi $ remain aligned through backpropagation, making the latter a valid and informative proxy for studying entropy’s influence on policy gradient magnitude.

---

### Decision · Program_Chairs · 2025-09-17

**Decision:**

Accept (poster)

**Comment:**

This paper first analyzes the existence of high-entropy minority tokens as forking tokens during the CoT reasoning of LLMs and observes that restricting policy gradient updates to these forking tokens during Reinforcement Learning with Verifiable Rewards (RLVR) improves mathematical reasoning performances of Qwen3-32B.

Overall, the entropy-based analysis and the corresponding simple modification of RLVR seem to be interesting and technically sound. Lack of experiments regarding different reasoning tasks and LLMs are somewhat (not fully) addressed during the rebuttal. Further analysis showing that RLVR with only 20% of high-entropy tokens leads to longer responses with more use of self-verification words is also interesting and helpful in supporting the claims of the paper. Based on the consensus of all positive ratings between the reviewers, I would recommend the paper to be accepted.

However, more empirical validation with diverse reasoning tasks and various LLMs would be necessary. Also, current theoretical analysis would be still insufficient. In particular, it remains unclear why updating only the influential tokens can directly lead to performance improvements. Moreover, the choice of 20% as the decision threshold is unlikely to be universal; from the perspective of the exploration–exploitation transition, a more thorough theoretical analysis on what constitutes an appropriate setting would be necessary.